# Identification of sporulation genes in *Bacillus anthracis* highlights similarities and significant differences with *Bacillus subtilis*

**Fernando H. Ramírez-Guadiana[1], Anna P. Brogan[1], Yuanchen Yu[1], Caroline Midonet[1], Joel W. Sher[1], Ernst W. Schmid[2], Ian J. Roney[1], David Z. Rudner** [1]*

**1** Department of Microbiology, Harvard Medical School, Boston, Massachusetts, United States of America, **2** Department of Biochemistry and Molecular Pharmacology, Harvard Medical School, Boston, Massachusetts, United States of America

* rudner@hms.harvard.edu

## Abstract

The molecular basis of endospore formation in the model gram-positive bacterium *Bacillus subtilis* has been investigated for over half a century. Here, using high throughput and classical genetic approaches, we performed a comparative analysis of sporulation in the human pathogen *Bacillus anthracis*. A transposon-sequencing screen identified >150 genes required for *B. anthracis* sporulation. As anticipated, many of the genes that are critical for sporulation in *B. subtilis* were also required for *B. anthracis* sporulation. However, we identified >50 genes that are important for sporulation in *B. anthracis* but not in *B. subtilis,* and 22 *B. anthracis* sporulation genes that are absent from the *B. subtilis* genome. To validate the hits from our screen, we generated an ordered transposon-mutant library using Knockout Sudoku. Cytological analysis of a subset of the canonical sporulation-defective mutants revealed similar but not identical phenotypes in the pathogen compared to the model. We investigated several of the newly identified sporulation genes, with an in-depth analysis of one, ORF *04167*, renamed *ipdA*. Sporulating cells lacking *ipdA* are blocked in the morphological process of engulfment, generating septal bulges. An AlphaFold-Multimer screen and a classical genetic enrichment revealed that IpdA is a secreted inhibitor of the polysaccharide deacetylase PdaN. Our data support a model in which induction of IpdA at the onset of sporulation inhibits deacetylation of the cell wall peptidoglycan (PG), enabling the sporulation-specific PG hydrolases to catalyze engulfment. Altogether, our studies reveal that *B. subtilis* is an excellent model for endospore formation in *B. anthracis*, while underscoring the importance of direct analysis in *B. anthracis*. The suite of tools that we have generated will catalyze the molecular dissection of sporulation and other cell biological processes in this important human pathogen.

**Data availability statement:** Raw Tn-seq sequence reads are available in the Sequence Read Archive (Accession: PRJNA1303624). Original and uncropped minimally adjusted images of the Western blot shown in the manuscript can be found in S1 Raw Images.

**Funding:** This work was supported by the National Institutes of Health (R01AI164647 and R21AI171308 to D.Z.R.; F31AI181098 to A.P.B.) and the National Science Foundation (DGE1745303 to A.P.B.). The funders had no role in study design, data collection and analysis, decision to publish, or preparation of the manuscript.

**Competing interests:** The authors have declared that no competing interests exist.

**Abbreviations:** AF-M, Alphafold-Multimer; BHI, brain-heart infusion; BKE, Bacillus knockout; CFUs, colony forming units; DSM, Difco Sporulation medium; gDNA, genomic DNA; GlcNac, N-acetyl-glucosamine; LB, Luria-Bertani; NBYG, nutrient broth/yeast extract/glycerol; PA, Phage Assay; pAE, predicted alignment error; PBST, PBS-0.05% Tween-20; PG, peptidoglycan; RMSD, root mean squared deviation; Tn-seq, transposon-insertion sequencing; TraSH, transposon site hybridization; WT, wild-type.

## Introduction

*Bacillus anthracis* endospores are the infectious biological entities that cause the zoonotic disease anthrax. Upon abrasion, ingestion, or inhalation, spores access subcutaneous tissues, gastrointestinal mucosa, or alveolar spaces where they are phagocytosed by macrophages, germinate, and cause cutaneous or systemic disease [1–3]. Because vegetative cells survive poorly in the environment, the cycle of germination, growth, and sporulation is critical to *B. anthracis* pathogenesis [4]. Although anthrax is often associated with bioterrorism, most human cases arise from contact with infected animals or contaminated animal products [1,2,5]. Environmental fluctuations due to climate change, deforestation, and agricultural encroachment have increased seasonal anthrax outbreaks in endemic areas and have led to the emergence of new zoonotic pathogens from the *Bacillus cereus* group, which includes *B. anthracis* [2,6–10]. A deeper understanding of the molecular mechanisms underlying *B. anthracis* sporulation and germination will facilitate the development of strategies for more effective disease prevention and treatment.

For over half a century, the nonpathogenic soil bacterium *Bacillus subtilis* has served as a model for sporulation and germination of endospore-forming pathogens like *B. anthracis*. Molecular, genetic, and cytological studies in *B. subtilis* have identified virtually all the genes required for these processes and defined the molecular basis for how this organism enters and exits dormancy (reviewed in [11–14]). By contrast, only a handful of sporulation and germination factors have been characterized in *B. anthracis* and other members of the *B. cereus* group [4]. Almost all of these are homologs of *B. subtilis* proteins. Phylogenetic studies suggest that *B. subtilis* and *B. anthracis* shared a common ancestor >800 million years ago [15], suggesting that there are likely to be similarities and differences between sporulation and germination in the two species. Here, we leveraged high-throughput genetics and modern molecular approaches to investigate how good a model *B. subtilis* has been for this important human pathogen.

Sporulation in *B. subtilis* and *B. anthracis* involves a set of highly orchestrated morphological processes. Upon nutrient limitation, both species initiate endospore formation [4,12]. The first landmark event in this process is an asymmetric division that generates a large cell (called the mother cell) and a smaller cell (the prospective spore or forespore). After polar division, the mother cell membranes migrate around the spore in a phagocytic-like process called engulfment, generating a cell within a cell. Membrane fission at the cell pole releases the forespore into the mother cell cytoplasm. The mother cell then assembles a set of protective layers around the forespore, while the prospective spore prepares for dormancy. Upon maturation, the mother cell lyses, releasing the spore into the environment.

This differentiation process takes ~7 hours in *B. subtilis* and requires >140 genes that are controlled by a series of stage- and compartment-specific transcription factors [13,16,17]. Although the mother cell and forespore follow distinct programs of gene expression they remain linked through signal transduction pathways. Entry into sporulation is governed by two transcriptional regulators: the response-regulator

Spo0A and the stationary phase sigma factor σ$^H$. The proteins produced under their control shift the cell division site from a medial to a polar position. These regulators are also responsible for the expression of compartment-specific transcription factors that are held inactive prior to polar division. Asymmetric division triggers the activation of the first cell-type-specific sigma factor σ$^F$ in the forespore. σ$^F$ controls the expression of several forespore proteins, including a secreted signaling molecule that activates the first mother cell transcription factor σ$^E$. Proteins produced under σ$^E$ control degrade the septal peptidoglycan that separates the two cells and promote the migration of the mother cell membranes around the forespore. During this process, a transenvelope complex is assembled in the membranes that separate the mother cell and forespore. This complex, which resembles a specialized secretion system, allows the mother to maintain forespore physiology and developmental gene expression [18–20]. After the completion of engulfment, a second forespore transcription factor σ$^G$ produces two signaling proteases that are secreted into the space between mother and forespore membranes where they trigger the activation of the mother cell transcription factor σ$^K$. A large set of proteins produced under the control of σ$^E$ and σ$^K$ in the mother cell is responsible for assembling a thick layer of spore-specific peptidoglycan between mother cell and forespore membranes and a multi-layered proteinaceous coat on the spore exterior [21]. In addition, σ$^G$ in the forespore controls the expression of proteins that prepare the spore for dormancy, including DNA-binding proteins that compact and protect the spore chromosome; a membrane complex that imports dipicolinic acid, facilitating spore dehydration and heat-resistance; and germinant receptors that enable the dormant spore to detect nutrient signals and trigger exit from dormancy [22,23]. Finally, a set of cell wall degrading enzymes produced in the mother cell under σ$^K$ control induce mother cell lysis and release of the mature spore.

Here, we performed a high-throughput sporulation screen in *B. anthracis* using transposon-insertion sequencing. Our screen identified over 150 potential sporulation genes, more than half of which encode homologs of *B. subtilis* sporulation factors. To validate the hits from the screen, we constructed an ordered transposon-mutant library that contains ~70% of the nonessential genes in *B. anthracis*. We analyzed a subset of Tn mutations that, in *B. subtilis,* impair polar division, engulfment, forespore and mother cell transcription, signaling between the two compartments, and maintenance of forespore physiology. In almost all cases, the *B. anthracis* mutants had similar although not identical morphological defects. We investigated a subset of the 50+ genes that were critical for sporulation in *B. anthracis* but not in *B. subtilis* and performed a thorough characterization of a gene that is unique to *B. anthracis*. The detailed characterization of this factor revealed that *B. anthracis*-specific sporulation factors modulate pathogen-specific growth and physiology in ways that facilitate efficient sporulation. Altogether, our findings indicate that *B. subtilis* is an excellent model for spore formation in *B. anthracis* and also highlight the importance of directed studies in the pathogen. The genetic and molecular tools established here will enable mechanistic dissection of *B. anthracis* sporulation, illuminate adaptations linked to pathogenesis, and enable deeper exploration of developmental processes across the *B. cereus* group.

## Results

### A screen for mutants defective in *B. anthracis* sporulation

To screen for sporulation-defective mutants by transposon-insertion sequencing (Tn-seq) [24], we built a plasmid-based transposon delivery system for *B. anthracis* modeled after one used previously for *B. subtilis* [25]. This plasmid harbors a temperature-sensitive replicon, a hyperactive allele of the mariner-Himar1 transposase, and a spectinomycin resistance cassette flanked by inverted repeats recognized by the transposase (see Materials and methods). An MmeI restriction site was engineered into one of the inverted repeats to aid in the preparation of the sequencing library. Using this system, we generated a library with >10$^6$ transposants in the plasmid-less Sterne strain 9131 and mapped the insertion sites by next generation sequencing (see Materials and methods). The library contained >170,000 unique insertions that were well distributed across the genome (Fig 1A), with a Tn insertion, on average, every 35 bp. Using EL-ARTIST [26] to analyze the results of this Tn-seq experiment, we identified essential genes in the parental strain. Of the 5,456 genes in the *B. anthracis* genome, 400 were predicted to be essential for growth in BHI medium (S1 Table).

PLOS Biology

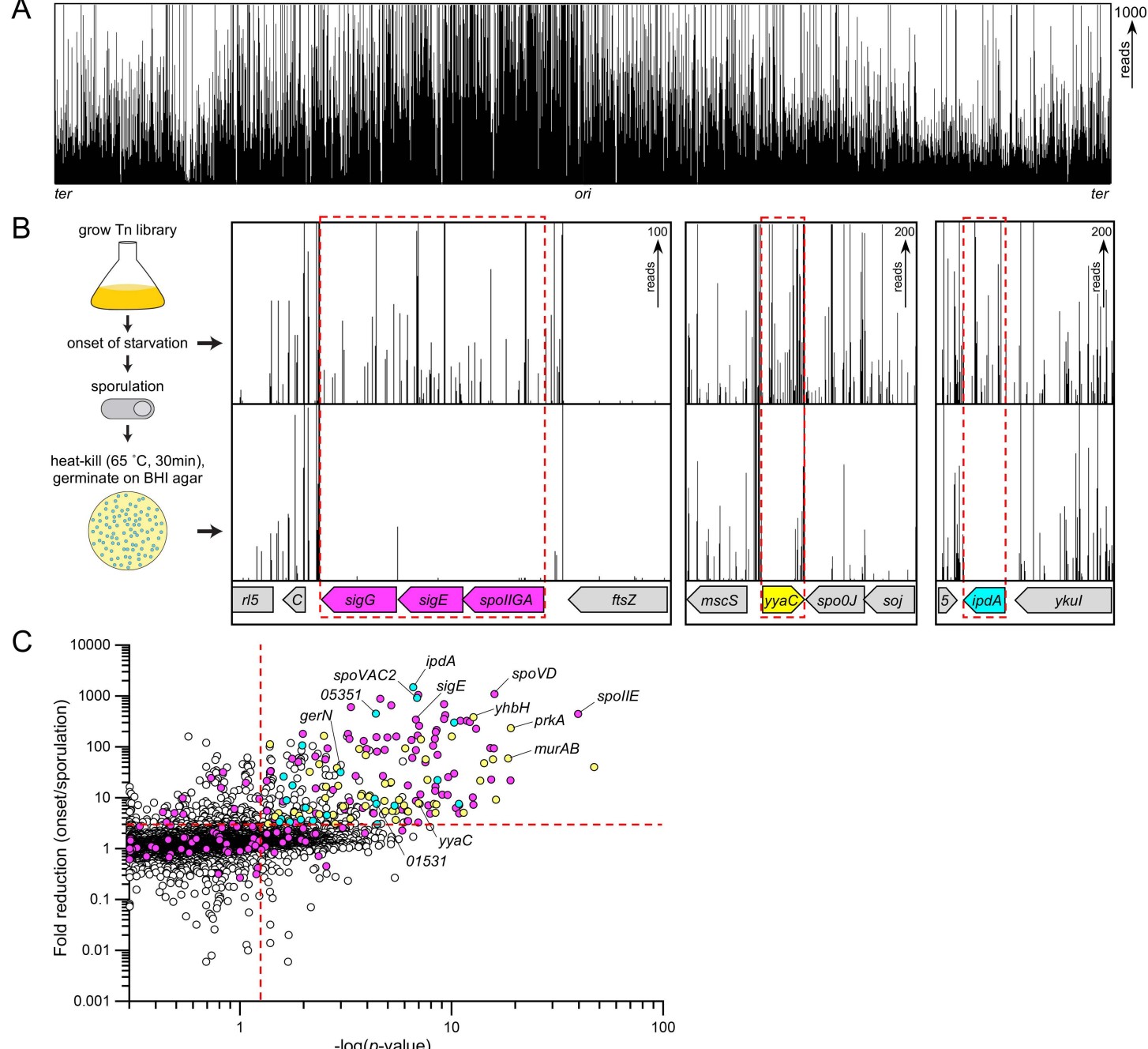

**Fig 1. A Tn-seq screen identifies a comprehensive set of genes required for sporulation in *B. anthracis*. (A)** Transposon-insertion profile across the entire *B. anthracis* genome of the initial transposon library. Each line indicates a transposon-insertion site in the genome and its height reflects the relative number of insertions mapped to that site. The maximum read height for panels (A) and (B) was set to the number shown on the right. **(B)** Schematic of the sporulation screen. The *B. anthracis* transposon library was grown in PA liquid medium. At the onset of starvation, a sample was removed, and the remaining culture was sporulated. The sporulated culture was incubated at 65 °C for 30 min to kill vegetative cells and sporulation-defective mutants. The surviving spores were germinated and outgrown on BHI agar plates and the resulting colonies pooled. Transposon-insertion sites and their abundance were determined by Tn-seq. Transposon-insertion profiles from three regions of the genome are depicted. Transposon-insertions in *sigG* (*spoIIIG*), *sigE* (*spoIIGB*), *spoIIGA*, *yyaC*, and *ipdA* were underrepresented after sporulation, germination, and outgrowth. Insertions in the essential *ftsZ* gene were not tolerated under either condition. **(C)** Scatterplot highlighting all the potential *B. anthracis* sporulation genes identified in the Tn-seq screen. Every gene in the *B. anthracis* genome is plotted on the graph. The Y-axis is the log fold-reduction in transposon-insertions at the onset of starvation

relative to the insertions after sporulation, heat treatment, germination, and outgrowth. The X-axis is the –log of the Mann–Whitney $U$ $p$-value. The statistical significance thresholds ($p \leq 0.05$, fold-reduction ≥3) are indicated by dashed red lines. *B. anthracis* genes that encode *B. subtilis* homologs that are required for efficient sporulation in *B. subtilis* are shown in magenta. *B. anthracis* genes that meet the statistical significance thresholds but are not required for sporulation in *B. subtilis* are highlighted in yellow. *B. anthracis* genes that meet the statistical thresholds for which there are no *B. subtilis* homologs are highlight in cyan. All candidate sporulation genes in *B. anthracis*, including those outside the statistical significance thresholds, are listed in S2–S6 Tables. The data underlying panel C can be found in S1 Data.

We used the Tn library to screen for genes required for spore formation, heat-resistance, germination, and outgrowth. The library was grown in Phage Assay (PA) medium and at the onset of starvation, an input sample was collected and the remaining culture was sporulated by nutrient exhaustion (Fig 1B). The culture was harvested seven hours after starvation, a time when Sterne strain 9131 produced the maximum number of heat-resistant spores (see Materials and methods). The culture was then incubated at 65 °C for 30 min to kill vegetative cells and sporulation-defective mutants. The spores were plated on BHI agar plates and ~$10^6$ colonies from the germination-proficient spores were pooled. Finally, the transposon-insertions at the onset of starvation and after sporulation, heat treatment, germination, and outgrowth (for simplicity, referred to as sporulation) were mapped by next generation sequencing. The location and number of reads for each transposon-insertion were compared between the input and output samples (Fig 1B and 1C). We identified >200 genes in which transposon-insertions were statistically (Mann–Whitney $U$; $p \leq 0.05$) underrepresented by ≥3-fold after sporulation compared to the onset of starvation (Fig 1C and S2–S6 Tables). Visual inspection of the transposon-insertion profiles revealed that ~22% of these genes had relatively few insertions at the onset of starvation and most of these were not considered further. In total, our screen identified 159 putative sporulation genes in *B. anthracis*.

## Similarities between *B. anthracis* and *B. subtilis* sporulation

In *B. subtilis*, ~142 genes are required for efficient sporulation [17]. Our Tn-seq screen identified 79 of these as putative sporulation genes in *B. anthracis* (magenta circles in Fig 1C and S2–S6 Tables). Among the 142 *B. subtilis* sporulation genes, 74 have "*spo*" names and 56 of these display strong sporulation defects (≤5% sporulation efficiency compared to wild-type (WT)) when mutated. Our *B. anthracis* sporulation Tn-seq screen identified 38 of these (S2 Table). Fig 1B shows the Tn-seq profile from a small region of the *B. anthracis* genome that contains three of these *spo* genes, *spoIIGA*, *spoIIGB* (*sigE*), and *spoIIIG* (*sigG*). Fig A in S1 Text contains Tn-seq profiles that highlight 11 additional *B. anthracis* *spo* genes. Of the 56 *B. subtilis* *spo* genes with strong sporulation defects, 18 were not identified in our *B. anthracis* Tn-seq screen. Two barely missed our cutoff (*spoIIAB*, *spoVM*) and two (*spo0A*, *spo0H*) had almost no insertions at the onset of sporulation. Eleven have paralogs in *B. anthracis* (e.g., *spoIIP*, *spoIIIJ*, and *spoIVH*) and might require the loss of both genes to impair sporulation. In addition, the *sigK* gene that encodes pro-SigmaK, is not interrupted by an intervening (*skin*) element in *B. anthracis* [27]. Accordingly, *B. anthracis* has a single *sigK* gene rather two gene fragments (*spoIVCB*, *spoIIIC*) and lacks *spoIVCA* that encodes the recombinase that excises the *skin* element during sporulation. This accounts for two more missing genes. The final gene was *spoIISB* which encodes an anti-toxin that inhibits the SpoIISA toxin during sporulation in *B. subtilis* [28]. Thus, SpoIISA is unlikely to impair sporulation in *B. anthracis*. We conclude that most of the core sporulation factors in *B. subtilis* are required for spore formation in *B. anthracis*.

*B. subtilis* has an additional 19 genes that do not have *spo* names but, when mutated, have strong sporulation defects (≤5% sporulation efficiency compared to WT). Among these, 10 were hits in our *B. anthracis* screen (S3 Table). Of the remaining 9, one barely missed the cutoff (*gerM*); one (*ricT*) had insertions exclusively at the 3′ end of the gene after sporulation; two (*ricA*, *ricF*) had virtually no insertions at the onset of sporulation; and three (*asnO*, *yqhT,* and *ccdA*) have putative paralogs. The remaining two genes (*fin* and *putB*) were unambiguously not underrepresented in the *B. anthracis* screen, suggestive of real differences in sporulation between *B. anthracis* and *B. subtilis*. Finally, similar to *B. subtilis*, a subset of the genes in the TCA cycle were hits in the *B. anthracis* screen, while several others were essential (S4 Table).

Altogether, these findings suggest that the core machinery that drives spore formation is hard-wired in both the model and pathogen. We note that 63 of the *B. subtilis* sporulation genes with more mild mutant phenotypes were not identified in our screen. Although some of these genes are not present in *B. anthracis* (*yerC*, *skfE,* and *skfF*), the failure to identify the others (e.g., *racA*, *sirA*, *fisB*, *spoVIGA*, and *spoVIGB*) suggests that our screen was not sensitive enough to detect them or that they are not critical for sporulation in *B. anthracis.*

## Differences between *B. anthracis* and *B. subtilis* sporulation

Our Tn-seq screen identified 58 putative sporulation genes in *B. anthracis* that are present in *B. subtilis* but have not been reported to be required for efficient sporulation in the model (yellow circles in Fig 1C and S5 Table). None of these genes were identified as hits in a previous *B. subtilis* sporulation Tn-seq screen [17]. Among the 58 genes, 25 of them are induced during *B. subtilis* sporulation but have modest or no reported phenotypes [16,29,30]. The identification of the *B. anthracis* homologs in our screen, underscores species-specific differences and highlights the importance of investigating sporulation in the pathogen. Fig 1B shows the Tn-seq profile from a small region of the *B. anthracis* genome that contains one of these *B. anthracis* genes, *yyaC*. Fig B in S1 Text compares Tn-seq profiles from regions of the *B. anthracis* and *B. subtilis* genomes that contain five hits (*yyaC*, *yqfT*, *yhfN*, *ytxC*, and *ysxE*) in the *B. anthracis* screen but not the *B. subtilis* screen.

Finally, our Tn-seq screen uncovered 22 putative sporulation gene in *B. anthracis* that are not present in the *B. subtilis* genome (cyan circles in Fig 1C and S6 Table). One such gene, locus *04176* (renamed *ipdA*), is illustrated in Fig 1B and two additional examples are shown in Fig C in S1 Text.

## Validation of hits from the screen

To experimentally validate hits from the sporulation screen, we constructed an ordered Tn mutant library using the Knock-out Sudoku method [31] (See Materials and methods). Briefly, ~52,000 transposants from our Tn library were arrayed in 96-well plates and saturated cultures from the arrayed mutants were distributed combinatorially into 47 distinct pools. These pools were sequenced and individual Tn mutants mapped to their unique well in the arrayed library based on the pools in which the mutant was identified. We then cherry-picked a mutant from all represented genes in the library to generate a condensed collection. This nonredundant quality-controlled collection contains 3,826 Tn mutants that represent ~70% of the 5,456 genes in the *B. anthracis* genome.

To test the hits from our screen, we transduced the Tn mutations into Sterne strain NR-9401. This strain sporulates asynchronously, but the total number of heat-resistant spores that can be recovered after 30 hours of nutrient exhaustion is approximately 10-fold higher and more reproducible than the plasmid-less strain we used to build the ordered mutant library (see Materials and methods). In total, we tested the sporulation efficiency of 71 hits from our screen. Of these, 62 were validated, with sporulation efficiencies ranging from <0.0001% to 40%. Forty-three of these had sporulation efficiencies of ≤5% compared to WT. The complete dataset can be found in S2–S6 Tables.

## Analysis of conserved sporulation genes

Our data indicate that homologs of many of the well-characterized *B. subtilis* sporulation genes are important for efficient sporulation in *B. anthracis* (S2 Table). To investigate whether these factors play similar roles in the morphological process of sporulation, we optimized conditions to visualize *B. anthracis* sporulation by fluorescence microscopy. Fig 2A shows a comparison of a sporulation time course of WT *B. subtilis* and the plasmid-less *B. anthracis* Sterne strain. Other than the striking difference in cell size, the morphological process of sporulation was similar between model and pathogen. To monitor gene expression in the mother cell and forespore compartments, we generated transcriptional reporters in which the *spoIIQ* promoter was fused to *yfp* and the *spoIID* promoter was fused to *cfp*. In *B. subtilis*, *spoIIQ* is expressed under the control of the first forespore-specific transcription factor $\sigma^F$ and *spoIID* is controlled by the mother cell-specific

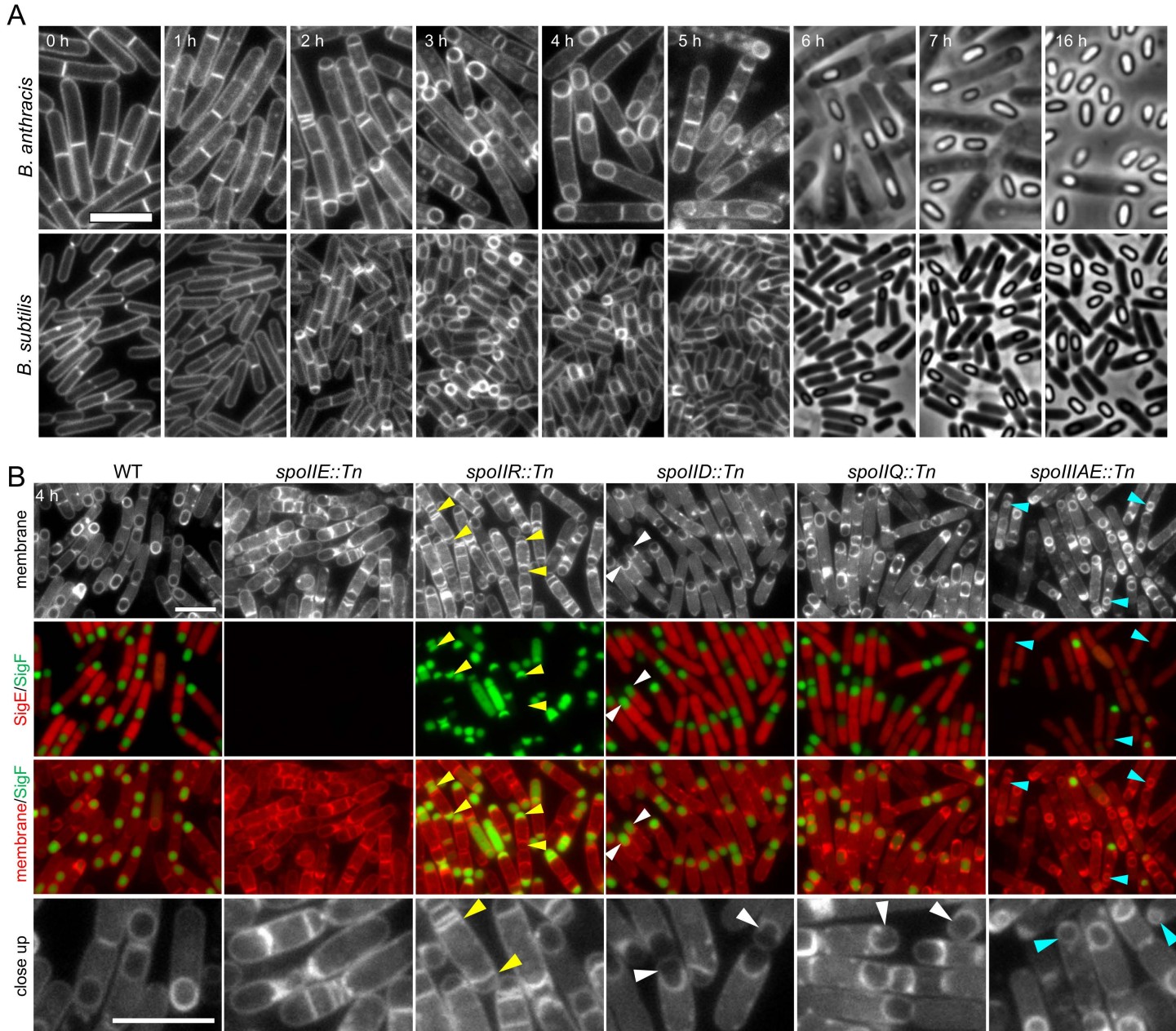

**Fig 2. Conserved *B. anthracis* sporulation genes have mutant phenotypes that are similar, but not identical, to *B. subtilis* mutants. (A)** Representative fluorescence and phase-contrast micrographs of sporulating *B. anthracis* and *B. subtilis*. Wild-type strains were induced to sporulate by nutrient exhaustion at 37 °C in PA (*B. anthracis*) or DSM (*B. subtilis*). Time after the onset of sporulation is indicated in the top left corner of the *B. anthracis* images. The membranes were stained (0–5 hour) with TMA-DPH. **(B)** Representative fluorescence images of the indicated *B. anthracis* mutants at hour 4 of sporulation. All strains harbor forespore (P*spoIIQ*-*yfp*) and mother cell (P*spoIID*-*cfp*) transcriptional reporters. σ^F activity in the forespore (SigF) and σ^E activity in the mother cell (SigE) were visualized with YFP (false-colored green) and CFP (false-colored red), respectively. The membranes were stained with TMA-DPH. Disporic cells in *spoIIR::Tn* (yellow carets), impaired engulfment in *spoIID::Tn* and *spoIIQ::Tn* (white carets), and small forespores in *spoIIIAE::Tn* (light blue carets) are highlighted. Scale bars indicate 5 μm. Images from a complete sporulation time course for each mutant can be found in Figs D and E in S1 Text.

transcription factor σ$^E$ [32,33]. As can be seen in Fig 2B and Fig D in S1 Text, the *B. anthracis* promoters have similar compartment-specific expression patterns to those in *B. subtilis*. We selected five genes (*spoIIE*, *spoIIR*, *spoIID*, *spoIIQ*, and *spoIIIAE*) that have been extensively characterized in *B. subtilis* and analyzed the *B. anthracis* Tn mutants during sporulation by fluorescence microscopy.

**spoIIE.** In *B. subtilis*, the dual-function membrane phosphatase SpoIIE promotes the shift from medial to polar division at the onset of sporulation and is required for the activation of σ$^F$ in the forespore after cytokinesis [34–36]. The failure to activate σ$^F$ in the Δ*spoIIE* mutant prevents the signal transduction pathway that activates σ$^E$ in the mother cell [37]. Similarly, the *B. anthracis spoIIE::Tn* mutant was delayed in polar division (Figs D and E in S1 Text) and failed to activate σ$^F$ in the forespore and σ$^E$ in the mother cell (Fig 2B). At later time points, the forespore compartments enlarged and the sporangia became wider with additional septa. These pleiotropic phenotypes have not been reported for the *B. subtilis* *spoIIE* mutant and hint at potential differences between the two species.

**spoIIR.** The *B. subtilis* SpoIIR protein is produced under σ$^F$ control in the forespore and functions as a secreted signal that activates σ$^E$ in the mother cell [38,39]. Sporulating cells lacking *spoIIR* fail to activate σ$^E$. In the absence of σ$^E$-directed gene expression, a second asymmetric septum is formed at the opposite cell pole, generating a disporic mutant [12]. The *B. anthracis spoIIR::Tn* mutant had similar phenotypes (Fig 2B and Fig D in S1 Text). In addition and similar to the *B. anthracis spoIIE::Tn* mutant, a subset of the developmentally arrested cells formed additional septa at late timepoints (Fig E in S1 Text).

**spoIID.** *B. subtilis spoIID* encodes a cell wall hydrolase required for the morphological process of engulfment [40–42]. After polar division, SpoIID is produced in the mother cell under the control of σ$^E$ [32]. The protein specifically localizes to the polar septum where it degrades the septal PG and promotes the movement of the mother cell membranes around the forespore [40,42]. Sporulating cells lacking SpoIID arrest at a very early stage of engulfment with a characteristic membrane bulge at the septal midpoint, due to the activity of a second PG hydrolase SpoIIP [43]. The *B. anthracis spoIID::Tn* was impaired in engulfment; however, unlike the *B. subtilis* mutant, engulfment was arrested when the mother cell membranes had migrated approximately halfway around the forespore (Fig 2B, Fig D and E in S1 Text). These findings suggest additional PG hydrolases in *B. anthracis* can initiate engulfment, but SpoIID-mediated cell wall cleavage is critical for the late stages of membrane migration.

**spoIIQ.** In *B. subtilis*, SpoIIQ has been implicated in several steps in spore morphogenesis. SpoIIQ is a bitopic membrane protein produced in the forespore under σ$^F$ control [33]. SpoIIQ specifically localizes to the septal membrane through interactions with the bitopic SpoIIIAH protein and the lipoprotein GerM that are both produced in the mother cell [44–46]. These transenvelope interactions promote the migration of the mother cell membrane around the forespore during engulfment [47]. Sporulating cells lacking *spoIIQ* have a modest but reproducible delay in engulfment. SpoIIQ-SpoIIIAH-GerM also reside in a larger transenvelope complex called the SpoIIQ-SpoIIIA complex that is required to maintain forespore development and is thought to function as a feeding tube that allows the mother cell to nurture the developing forespore [18–20,48]. In the absence of SpoIIQ, *B. subtilis* sporulating cells fail to assemble this complex, shrink in size, and have large membrane invaginations [48]. They also fail to maintain forespore gene expression. The mutant phenotype of *B. anthracis spoIIQ::Tn* was more pleiotropic (Fig 2B, Fig D and E in S1 Text). The early stages of engulfment looked similar to WT with modest delays. However, at later time points the forespores appeared heterogenous with a subset arrested prior to the completion of engulfment. Finally, the forespores at this late time point had membrane blebs but did not appear to shrink. We contrast these phenotypes with the *spoIIIAE::Tn* mutant phenotype described below.

**spoIIIAE.** *spoIIIAE* encodes one of the proteins in the SpoIIQ-SpoIIIA complex [48]. In *B. subtilis,* sporulating cells lacking *spoIIIAE* have a similar phenotype to the *spoIIQ* mutant: the engulfed forespores shrink, the spore membranes develop large invagination, and forespore gene expression cannot be sustained [18]. The *B. anthracis spoIIIAE::Tn* mutant had similar yet distinct phenotypes. Engulfment in the mutant was similar to WT; however, in most cases,

the forespores barely grew in size (Fig 2B, Fig D and E in S1 Text). The diminutive forespores lacked the membrane invaginations observed in the *B. subtilis spoIIIAE* mutant. Although the *spoIIIAE::Tn* mutant activated σ^F in the forespore and σ^E in the mother cell at early stages in sporulation, by hour 4, the σ^F activity reporter (YFP) was no longer detected in most forespore compartments (Fig 2B, Fig D in S1 Text). A loss of metabolic potential in the forespore could impair ongoing synthesis of YFP, but cannot explain the loss of fluorescence from the existing pool. Our findings suggest that this normally stable protein unfolds or is degraded in the *spoIIIAE::Tn* mutant. Altogether, these phenotypes support the idea that the SpoIIIA proteins are required to maintain the metabolic potential of the forespore and suggest that the loss of potential is experienced by the *B. anthracis* forespore at an even earlier stage of development. These data also indicate that SpoIIQ is not as critical for this process in *B. anthracis,* which is reminiscent of the modest phenotypes observed in sporulating *B. subtilis* lacking SpoIIQ's interaction partner SpoIIIAH [48].

In summary, our analysis indicates that the well-characterized sporulation factors in *B. subtilis* play similar but, in several cases, not identical roles in *B. anthracis*. These findings highlight the power of *B. subtilis* as a model and the importance of follow-up studies in the pathogen.

## Sporulation genes that are important in *B. anthracis* but are not critical in *B. subtilis*

Fifty-eight of the *B. anthracis* genes that were hits in our sporulation screen have *B. subtilis* homologs that are not required for efficient sporulation in the model. We chose seven (*yyaC, yhfN, yqfT, yuzB, ysxE, ytxC,* and *murAB*) to characterize further. We focused on genes that are induced during sporulation in *B. subtilis* [16,30] but have not been reported to have mutant phenotypes. We anticipated that the *B. anthracis* Tn mutants might have interesting morphological defects, and we liked the idea that studies on the pathogen could inform the model.

The *B. subtilis yyaC* gene is expressed under the control of σ^F in the forespore and encodes a soluble protein that is predicted to be in the peptidase HybD-like domain superfamily (IPR023430). Foldseek [49] predicts that YyaC is structurally similar to the *B. subtilis* germination protease Gpr that is expressed at a later stage in sporulation [50]. *B. anthracis* has a true Gpr homolog, suggesting that, if YyaC is a protease, it has a different function. Analysis of the *yyaC::Tn* mutant by fluorescence microscopy revealed that at a late stage in sporulation the forespores become shrunken with large membrane invaginations (Fig 3 and Fig F*a* in S1 Text). These defects are similar to the "collapsed forespore" phenotype in *B. subtilis spoIIQ* and *spoIIIA* mutants [48] and raise the possibility that YyaC-mediated degradation of forespore proteins is critical to maintain metabolic potential of the developing spore. We also observed increased cell lysis at late stages of sporulation in the *yyaC::Tn* mutant (Fig 3), suggesting that loss of forespore metabolic potential leads to mother cell lysis.

A second mutant, *yqfT::Tn*, also had collapsed forespores. However, these forespores had abnormal and heterogeneous morphologies and appeared to lose the ability to maintain compartmentalization (Fig 3 and Fig F*a* in S1 Text). At late stages of differentiation, the σ^F and σ^E fluorescent reporters were present in both compartments of the sporangia. The *B. subtilis yqfT* gene is expressed under σ^E control and encodes a soluble protein of unknown function. How this mother cell protein supports forespore development in *B. anthracis* remains to be discovered.

One of the *B. anthracis* mutants, *yuzB::Tn*, was impaired in entry into sporulation (Fig F*a* in S1 Text) and 4 hours after nutrient exhaustion only a small subset of cells had polar septa (Fig 3). *yuzB* is not expressed during sporulation in *B. subtilis* and encodes a small soluble protein. Foldseek predicts that YuzB has a thioredoxin fold, suggesting this protein could be involved in maintaining the cellular redox state as *B. anthracis* exhausts its nutrients. Two of the mutants we analyzed, *yhfN::Tn* and *ytxC::Tn,* produce dormant spores that were more spherical than WT (Fig 3 and F*b* in S1 Text). Aside from this subtle phenotype, the mutants appeared to undergo differentiation normally. Several of the mutants, exemplified by *ysxE::Tn*, produced normal-looking spores yet had a strong sporulation defect (Fig 3 and F*b* in S1 Text). These mutant spores fail to mature and could have defects in DPA accumulation, core dehydration, or germination.

Finally, we analyzed the *B. anthracis murAB::Tn* mutant. MurAB and its essential paralog MurAA catalyze the first committing step in PG precursor synthesis [51]. A recent *B. subtilis* study, using a sensitized background that

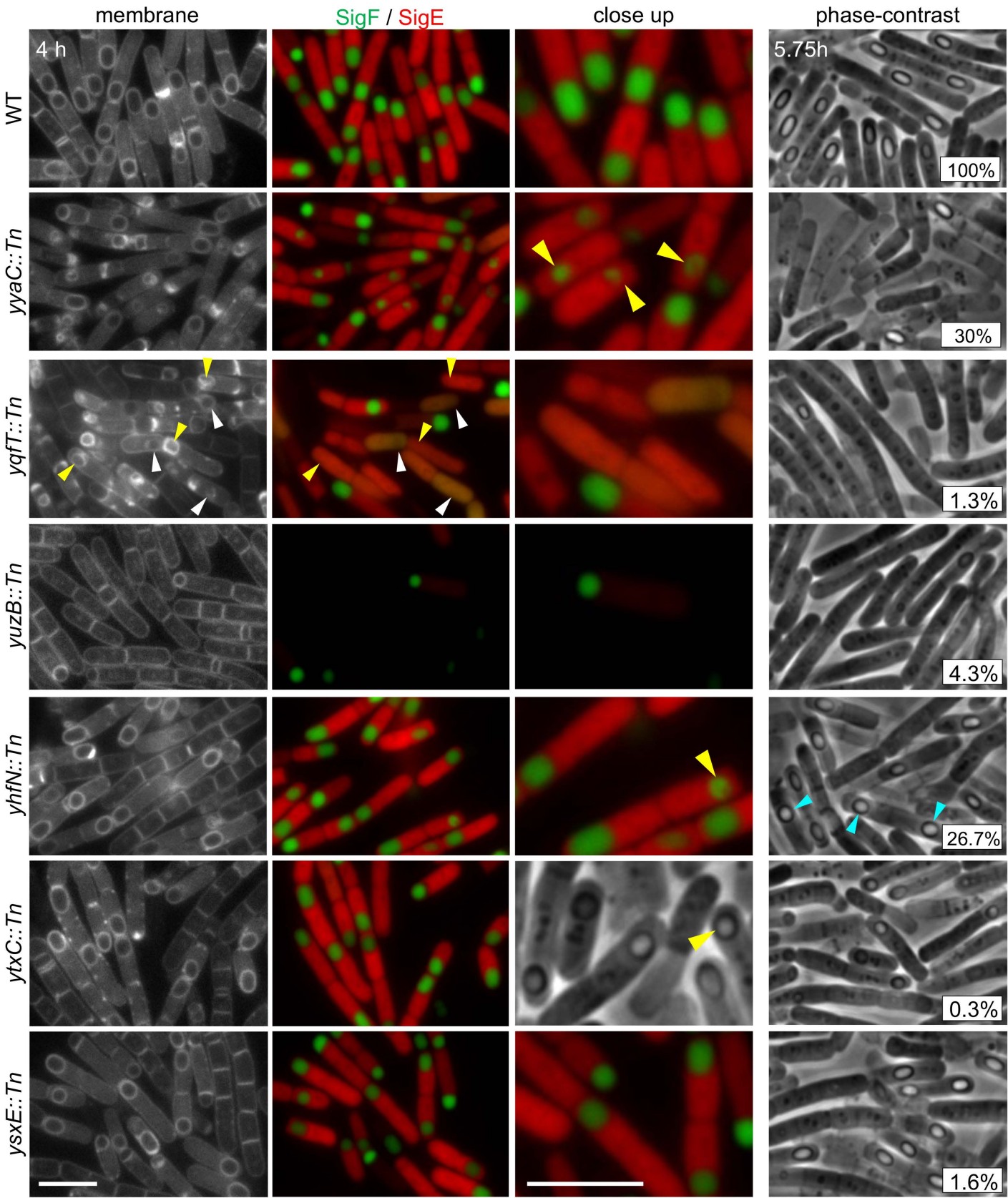

**Fig 3. Genes that are important for sporulation in *B. anthracis* but are not critical for spore formation in *B. subtilis*.** Representative fluorescence and phase-contrast images of the indicated *B. anthracis* mutants at hour 4 of sporulation. All strains have forespore and mother cell transcriptional reporters. The membranes were visualized with TMA-DPH. Phase-contrast images were from hour 5.75 of sporulation. The sporulation efficiency of each mutant, determined in the BaR6 background, is shown in the bottom right corner of the phase-contrast images. Collapsed or distorted forespores (yellow carets), spherical forespores (light blue carets) are highlighted. White carets highlight sporangia that have CFP and YFP in both compartments. Scale bars indicate 5 μm.

is partially impaired in engulfment, reported a more pronounced engulfment defect when *murAB* was also deleted [52]. However, *B. subtilis* sporulating cells that only lack *murAB* had no phenotype. Interestingly, the *B. anthracis* *murAB::Tn* mutant was severely impaired in sporulation (Fig G in S1 Text). The nutrient-exhausted cells had morphological defects at early stages of sporulation, followed by impaired polar division, defects in engulfment, and spore lysis (Fig G in S1 Text). These data suggest that MurAB is more critical for cell wall synthesis during starvation and sporulation and could reflect a greater division of labor between the two paralogs in *B. anthracis* compared to *B. subtilis*.

Altogether, our initial characterization of the *B. anthracis* sporulation genes that have *B. subtilis* homologs with weak or undetectable mutant phenotypes indicates that there are several additional factors required for sporulation in *B. anthracis*. Since 25 of these genes are induced during sporulation in *B. subtilis*, our findings suggest that some of these factors may play similar roles in *B. subtilis,* but alternative genes or pathways exist that compensate for their absence. Synthetic lethal sporulation screens with the *B. subtilis* mutants could be a fruitful path to revealing their function. Alternatively, enrichment screens in *B. anthracis* (see below) could guide functional characterization.

## 04176 (IpdA) is required for the morphological process of engulfment

Finally, we turned out attention to a putative sporulation factor 04176 that is unique to *B. anthracis*. Based on the experiments described below, we have renamed this protein IpdA for Inhibitor of polysaccharide deacetylases. The *ipdA::Tn* mutant is severely impaired in sporulation (<0.01% of WT) (Fig 4B and S6 Table) and fluorescence microscopy indicates that *ipdA::Tn* mutant cells are engulfment-defective and have septal membrane bulges that protrude into the mother cell cytosol (Fig 4A and Fig H in S1 Text). This phenotype is reminiscent of *B. subtilis* mutants lacking the engulfment cell wall hydrolases SpoIID (Fig 4C) or SpoIIP [41,42,53]. Importantly, expression of *ipdA in trans* restored efficient sporulation and normal engulfment to the *ipdA::Tn* mutant, confirming that the defects were attributable to the loss of IpdA (Fig 4A and 4B).

IpdA is a small soluble protein with a predicted signal peptide. It does not have a *B. subtilis* homolog based on a BLAST homology search. However, Foldseek predicts that IpdA is structurally similar to *B. subtilis* IseA (Fig 4D), a secreted inhibitor of cell wall hydrolases that contain NlpC/P60 catalytic domains [54–56]. Neither IseA nor the NlpC/P60 peptidoglycan hydrolases have been implicated in sporulation in *B. subtilis* (Fig I*a* in S1 Text) and our analysis of an *iseA* mutant indicates that sporulation is unaffected in its absence (Fig 4C and Fig I*b* in S1 Text).

To investigate whether *B. anthracis ipdA* is induced during sporulation, we fused its promoter to *yfp* and monitored fluorescence during a sporulation time course. As can be seen in Fig 4E, *ipdA* was induced at an early stage of sporulation, prior to asymmetric division (Fig 4E and Fig J in S1 Text). Consistent with the idea that *ipdA* is controlled by the response-regulator Spo0A, its promoter region contains four near-consensus Spo0A binding motifs [57]. By contrast, analysis of an *iseA* transcriptional reporter revealed that it is not specifically induced during *B. subtilis* sporulation (Fig J in S1 Text), consistent with our findings that it is not required for successful sporulation.

*B. subtilis* IseA inhibits NlpC/P60 cell wall hydrolases by binding the catalytic groove of these enzymes (Fig K*a* in S1 Text) [54]. Importantly, several of these enzymes are required for cell separation after cytokinesis. Based on the

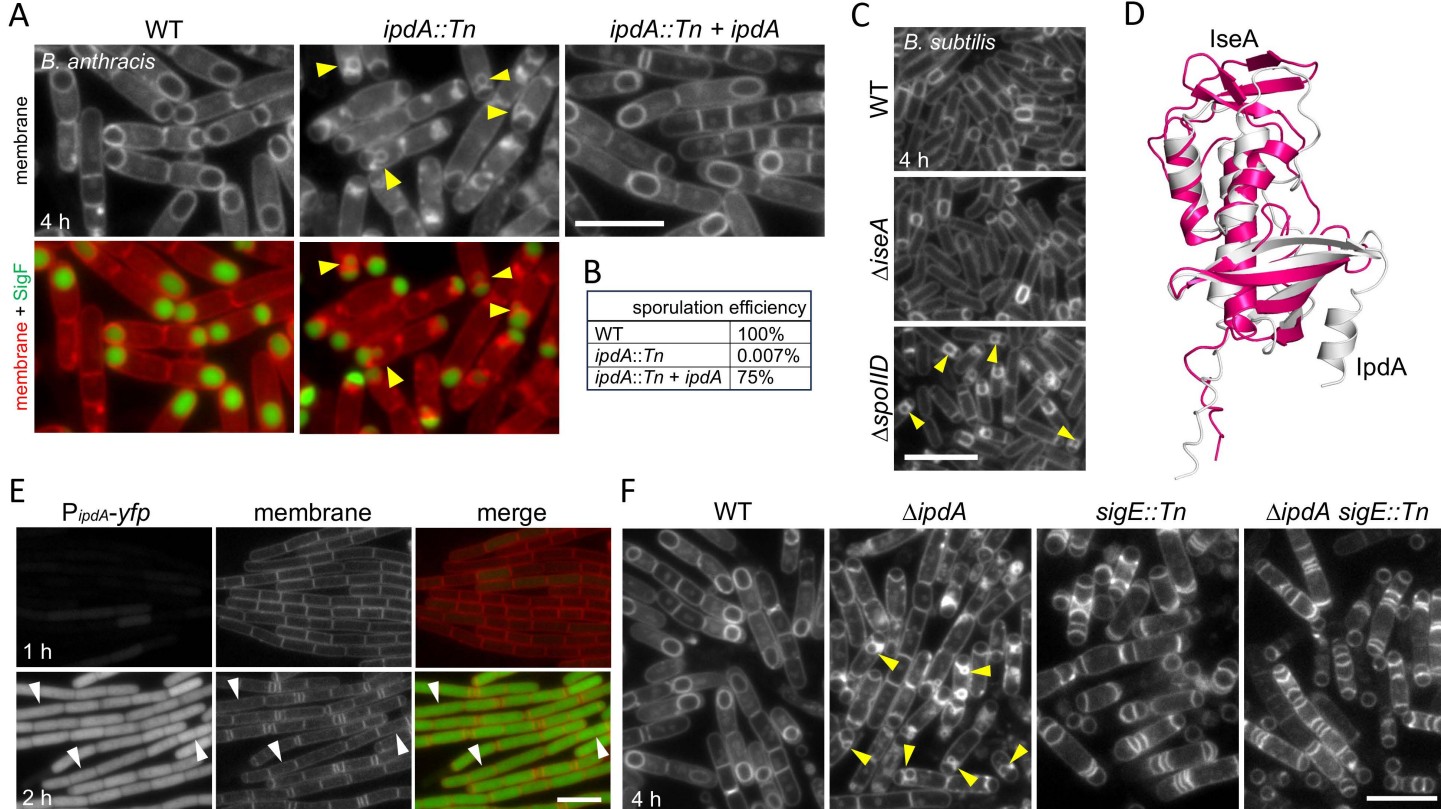

**Fig 4. IpdA (04176) is required for the morphological process of engulfment. (A)** Representative fluorescence images of the indicated *B. anthracis* strains at hour 4 of sporulation. Wild-type (WT) and *ipdA::Tn* harbor a σ$^F$-responsive promoter (P$_{spoIIQ}$) fused to *yfp* (false-colored green). The complementation strain (*ipdA::Tn + ipdA*) lacks the transcriptional reporter. The membranes were stained with TMA-DPH. Yellow carets highlight bulged septal membranes in the *ipdA::Tn* mutant. **(B)** Sporulation efficiency of the indicated strains in the BaR6 background. **(C)** Representative fluorescence images of the indicated *B. subtilis* strains at hour 4 of sporulation. Bulged septal membranes in the *ΔspoIID* mutant are highlighted (yellow carets). A complete sporulation time course of the *B. subtilis* strains can be found in Fig I in S1 Text. **(D)** Superimposition of the AlphaFold-predicted structure of IpdA (white) and the experimentally determined structure of IseA (magenta) (PDB:2RSX). The root mean squared displacement (RMSD) is 3.9 Å. **(E)** Representative fluorescence images of sporulating *B. anthracis* cells harboring an *ipdA* transcriptional reporter (P$_{ipdA}$-*yfp*). Time after the initiation of sporulation is indicated in the bottom left corners. Membranes were stained with TMA-DPA. White carets highlight cells with YFP fluorescence prior to polar division. YFP images were scaled identically. A complete time course can be found in Fig J in S1 Text. **(F)** Representative images of the indicated *B. anthracis* strains at hour 4 of sporulation. Yellow carets highlight septal bulges in the *ΔipdA* mutant. A complete time course can be found in Fig H in S1 Text. Scale bars indicate 5 μm. The data underlying panel B can be found in S1 Data.

structural similarity between IseA and IpdA, our initial hypothesis was that IpdA also inhibits PG hydrolases that function in cell separation during vegetative growth. In this model, the absence of IpdA leads to inappropriate septal PG cleavage after polar division resulting in septal bulges. To investigate this idea, we analyzed sporulating cells lacking the mother cell transcription factor σ$^E$ in the presence and absence of IpdA. In *B. subtilis*, σ$^E$ controls the expression of the engulfment cell wall hydrolases SpoIID and SpoIIP [43]. If IpdA inhibits vegetative cell wall hydrolases then sporulating cells lacking σ$^E$ and IpdA should have septal bulges, while the matched *ipdA+* cells should not. As can be seen in Fig 4F and Fig H in S1 Text, no septal bulges were observed in the *sigE::Tn* mutant in the presence or absence of IpdA. These findings suggest that IpdA does not inhibit the vegetative PG hydrolases and instead is required to promote or enable the cell wall degrading activities of the engulfment enzymes. Consistent with the idea that IpdA does not inhibit NlpC/P60 cell separases, AlphaFold-multimer (AF-M) did not predict interactions between IpdA and any of the *B. anthracis* NlpC/P60-containing PG hydrolases (Fig K*b* in S1 Text).

## Evidence that IpdA inhibits the polysaccharide deacetylase PdaN to promote engulfment

To investigate IpdA function, we performed a one-by-all AF-M screen, in which IpdA was folded pairwise with each protein in the *B. anthracis* proteome. Five independent models were generated for each prediction and the confidence metrics were analyzed using a published pipeline [58]. An average model score was assigned to each prediction based on how well the predicted contacts agreed across the five models, and the average predicted alignment error (pAE) across the contact region in the best model was measured (S7 Table). Hits with an average model score ≥3 and an average pAE ≤ 5 Å were considered strong candidates. Forty proteins met these criteria (Fig 5A) of which 12 were predicted to be secreted or membrane anchored (S7 Table). In a complementary approach, we performed a classical genetic enrichment for suppressors of the Δ*ipdA* mutant. Four independent cultures of the mutant were subjected to cycles of sporulation, heat-kill, and reinoculation into sporulation medium. After 3–5 cycles, the sporulation efficiency of each culture increased >1000-fold. A single colony from each was re-tested (Fig 5B) and the suppressor mutations identified by whole–genome sequencing. Three of the suppressors mapped to gene *04184* and one mapped to gene *02960* (Fig 5C). The 04184 protein is a member of the YkyA protein family (PF10368) that is annotated as a putative cell wall binding lipoprotein, although we could not find experimental evidence for PG binding. Based on our characterization of *04184*, we have renamed the gene *pdcF* for polysaccharide deacetylase co-factor. 02690 is homologous to a putative secreted polysaccharide deacetylase YheN (or PdaN) in *B. subtilis* [59] and we used the PdaN nomenclature for 02690 (Fig 5C). Previous biochemical studies of *B. anthracis* PdaN indicate that it can deacetylate N-acetyl-glucosamine (GlcNac) in vitro [60]. Importantly, PdaN was among the strongest hits in our AF-M screen (Fig 5A). Furthermore, three other polysaccharide deacetylases (00373, 05406, and 02021) [61] were also strong hits (Fig 5A). We validated *pdcF* and *pdaN* as suppressors of Δ*ipdA* using the Tn-insertion mutants (Fig 5B). Both loss-of-function mutants improved sporulation efficiency by >1000-fold.

Fig 5D shows the high-confidence AlphaFold3-predicted IpdA-PdaN complex. Importantly, residues in a short α-helix of IpdA are predicted to bind and block PdaN's catalytic pocket. Similar complexes are predicted for the other three deacetylases (Fig L in S1 Text). PdcF (04184) is predicted to be a lipoprotein with a four-helix bundle. Strikingly, AlphaFold3 (AF3) predicts a high-confidence interaction between PdaN and PdcF (Fig 5E and Fig M*a* and N in S1 Text), suggesting the two proteins function as a complex and may require each other in vivo for proper deacetylase activity. Importantly, AF3 predicts that IpdA can bind and inhibit PdaN in the context of the PdcF-PdaN complex (Fig 5F and Fig M*b* in S1 Text). Transcriptional reporters in which the *pdaN* and *pdcF* promoters were fused to *yfp* indicate that both factors are expressed during vegetative growth (Fig O in S1 Text), consistent with the model that they act together to modify the envelope during growth. In support of this idea, expression of both genes, but not either one alone, in *B. subtilis* caused impaired growth (Fig P*a* in S1 Text). Furthermore, fluorescence imaging revealed that co-expression of *pdaN* and *pdcF* specifically impaired cell elongation, resulting in short chubby cells (Fig 5G and Fig P*b* in S1 Text). These morphological defects are strikingly similar to the phenotypes associated with *B. subtilis* cells depleted of the PG hydrolases CwlO and LytE required for cell wall elongation [62,63]. Collectively, these findings argue that PdaN and PdcF act together to deacetylate peptidoglycan, blocking cell wall cleavage.

Although the *pdaN::Tn* mutant significantly suppressed the sporulation defect of Δ*ipdA* (Fig 5B), it only modestly suppressed the engulfment defect (Fig 5H). By contrast, *pdcF::Tn* was more effective at suppressing the septal bulges in the Δ*ipdA* mutant. These data raise the possibility that additional deacetylases are inhibited by IpdA and these enzymes might impair engulfment in the Δ*ipdA pdaN::Tn* background (see Discussion). Similarly, our findings also suggest that PdcF is required for the activity or membrane localization of one or several additional deacetylases. In support of this latter model, vegetative *B. anthracis* cells lacking *pdcF* but not *pdaN* are more susceptible to lysozyme than WT (Fig Q in S1 Text). Since deacetylation provides high-level lysozyme resistance in *B. anthracis* [64], this result argues that PdcF functions as a co-factor for several deacetylases and in its absence the cell wall is more acetylated and susceptible to lysozyme.

To investigate whether IpdA inhibits PdaN as predicted by the structural prediction, we substituted alanines for the three residues (N73, Q76, and D80) predicted to bind and occlude PdaN's catalytic pocket. Consistent with the AF3 prediction,

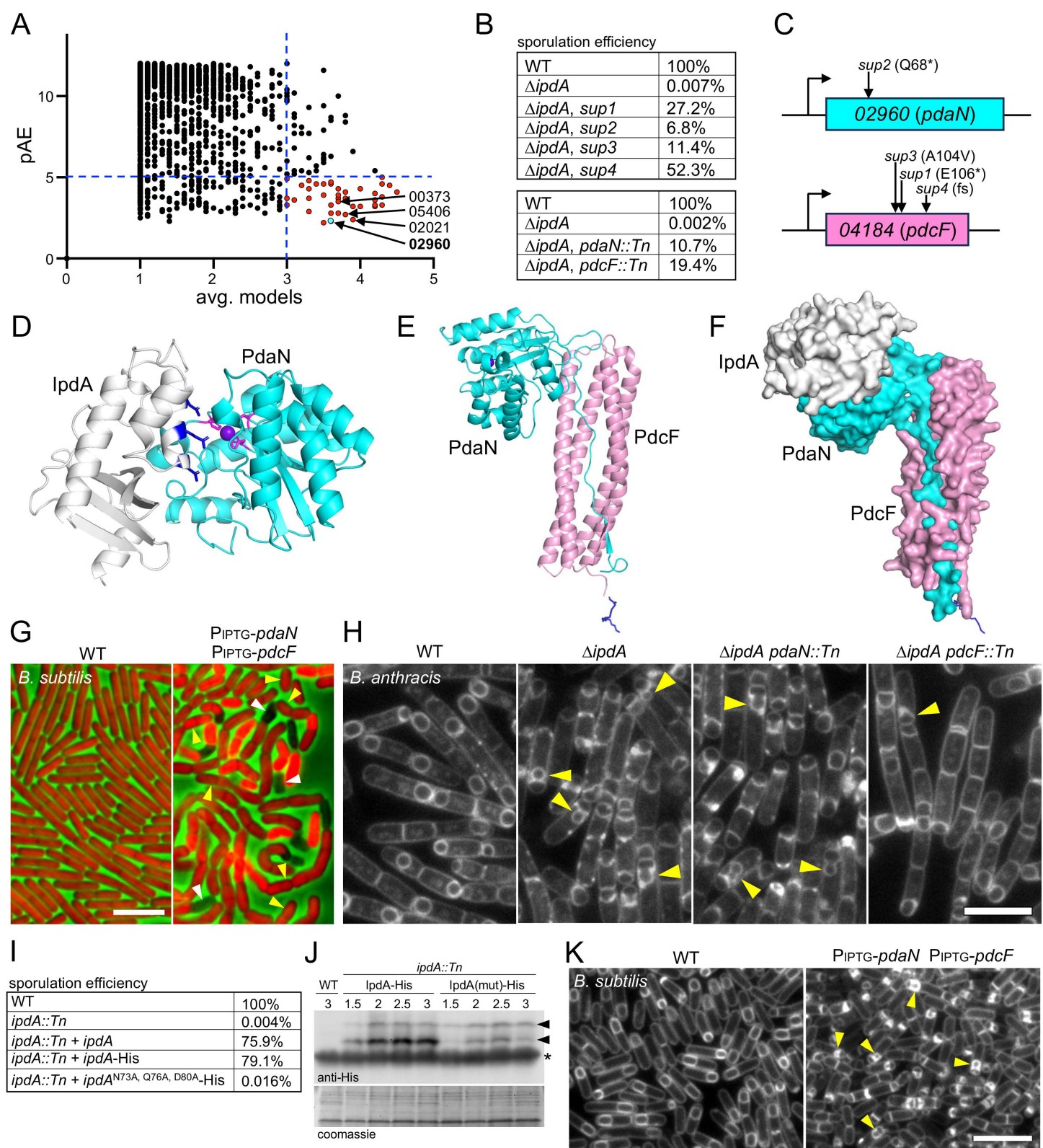

**Fig 5. IpdA inhibits the polysaccharide deacetylase PdaN (02960). (A)** Scatterplot of all potential hits (circles) from a one-by-all AlphaFold2-Multimer screen using IpdA as the bait. Five models were predicted for each protein in the *B. anthracis* proteome. All proteins with an average model score ≥1

are plotted against the average pAE score from the best model. Dashed blue lines show arbitrarily assigned significance thresholds (pAE ≤ 5, average models ≥3). Forty-four proteins met these criteria. 02960 (PdaN) is shown in cyan. Three other polysaccharide deacetylases are also indicated. **(B)** Sporulation efficiency of suppressors of Δ*ipdA* (top) identified through an enrichment screen (see Materials and methods). Validation of the hits from the screen using transposon-insertion mutants (bottom). **(C)** Schematics of the *02960* (*pdaN*) and *04184* (*pdcF*) genes with suppressor mutations highlighted. **(D)** AlphaFold3-predicted complex of IpdA (white) and the catalytic domain of PdaN (cyan). PdaN's catalytic residues (magenta) and $Zn^{2+}$ ligand (purple sphere) are shown. Residues in IpdA predicted to interact with PdaN are shown in dark blue. **(E)** AlphaFold3-predicted complex of PdaN (cyan) and PdcF (pink). The predicted lipidated N-terminus of PdcF is shown in dark blue. **(F)** AlphaFold3-predicted complex of IpdA (white), PdaN (cyan), and PdcF (pink). The pAE plots for Fig 5D–5F can be found in Figs L and M in S1 Text. **(G)** Representative fluorescence images of the indicated *B. subtilis* strains with cytoplasmic mCherry (red) 3 hour after addition of IPTG (500 µM, final) to induce expression of *B. anthracis pdaN* and *pdcF*. Short, chubby cells (yellow carets) that ultimately lyse (white carets) are highlighted. A complete time course can be found in Fig P in S1 Text. **(H)** Representative fluorescence images of the indicated *B. anthracis* strains at hour 4 of sporulation. Yellow carets highlight sporulating cells with septal bulges. *pdaN::Tn* modestly suppresses the morphological defects associated with Δ*ipdA* despite the 500-fold suppression of the sporulation defect. *pdcF::Tn* suppresses the septal bulging to a greater extent. **(I)** Sporulation efficiencies of the indicated strains. An IpdA variant with three amino acid substitutions in residues predicted to interact with PdaN fails to complement the Δ*ipdA* mutant. **(J)** Representative anti-His immunoblot of the indicated strains over a sporulation time course. Time in hours after the onset of sporulation is indicated above the blots. Black carets highlight the bands specific to IpdA-His. The lower band (highlighted with an asterisk) is lysozyme, used to generate the lysates. A Coomassie-stained gel was used to control for loading. **(K)** Representative fluorescence images of the indicated *B. subtilis* strains at hour 4 of sporulation. IPTG (500 µM, final) was added to the cultures at the onset of sporulation. Yellow carets highlight septal bulges. A complete time course can be found in Fig R in S1 Text. Scale bars indicate 5 µm. The data underlying panels A, B, and I can be found in S1 Data.

this protein variant failed to rescue the Δ*ipdA* mutant (Fig 5I). Immunoblot analysis of IpdA-His during a sporulation time course revealed that the levels of the IpdA variant were reduced relative to the WT protein (Fig 5J). The reduced protein levels could be due to instability caused by the amino acid substitutions or could result from a failure to interact with PdaN, leading to release of the mutant protein into the medium. We tentatively conclude that IpdA binds and inhibits PdaN in a manner predicted by AlphaFold.

Altogether, our data are consistent with a model in which induction of IpdA at the onset of sporulation inhibits PdaN and possibly other vegetative deacetylases. The acetylated PG sugars ensure that the sporulation-specific cell wall hydrolases can efficiently cleave the septal PG and promote engulfment. To further investigate this model, we co-expressed PdaN and PdcF in *B. subtilis* cells at the onset of sporulation and visualized the engulfment process over time. As can be seen in Fig 5K and Fig R in S1 Text, engulfment was impaired and many cells had septal bulges similar to those observed in the *B. anthracis* Δ*ipdA* mutant. Importantly, the engulfment defects required expression of both proteins (Fig R in S1 Text). We observed similar septal bulges in sporulating *B. subtilis* cell when we induced the expression of *B. subtilis pdaC* at the onset of sporulation (Fig R in S1 Text). PdaC is a membrane anchored MurNAc deacetylase that is not normally produced during sporulation [25,65]. We conclude that one or both of the engulfment hydrolases in *B. subtilis* require acetylated peptidoglycan to degrade the septal cell wall. We infer that the *B. anthracis* enzymes require a similar substrate and that induction of IpdA at the onset of sporulation ensures the septal PG remains sufficiently acetylated (Fig 6) (See Discussion).

## Discussion

Altogether, our Tn-seq screen identified 159 genes as potential sporulation factors in *B. anthracis*. Validation using our ordered transposon-mutant collection confirmed 87% of the tested hits, suggesting ~139 of the identified factors are required for efficient sporulation. Among these, we identified 79 genes that have *B. subtilis* homologs required for sporulation; >50 conserved genes that, in *B. subtilis*, are not critical for spore heat-resistance although several are induced during sporulation; and >20 *B. anthracis* genes required for sporulation that are not present in *B. subtilis*. The full dataset, the ordered mutant collection, and the tools we generated, provide a strong foundation for follow-up studies on these newly identified sporulation factors.

An important limitation of our study arises from the use of heat-resistance as the sole criterion for successful spore formation. Sporulation genes that contribute to other spore resistance properties were likely missed. For

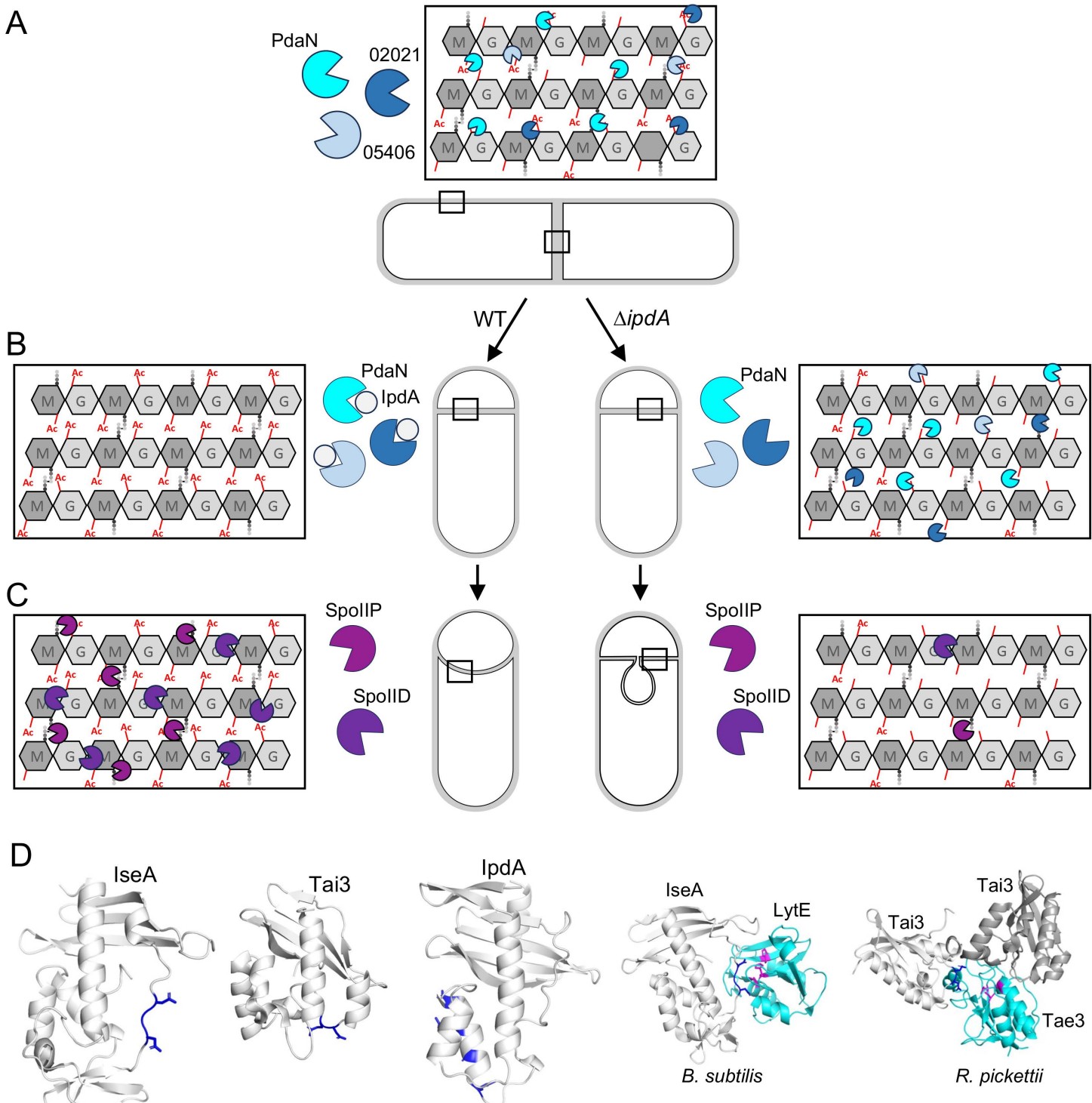

**Fig 6. Schematic model for IpdA function during sporulation in *B. anthracis*. (A)** During vegetative growth, PdaN, 02021, and 05406 are proposed to deacetylate the N-linked acetyl groups from GlcNAc (G) and MurNAc (M) sugars of the peptidoglycan (PG). **(B)** At the onset of sporulation but prior to polar division, IpdA is induced and inhibits deacetylase activity, resulting in acetylated septal PG (left). In the Δ*ipdA* mutant, the septal PG is deacetylated (right). **(C)** The cell wall hydrolases SpoIID and SpoIIP can cleave acetylated PG and promote engulfment (left). SpoIID and SpoIIP are unable to efficiently cleave deacetylated PG in the Δ*ipdA* mutant, resulting in septal membrane bulges (right). For simplicity, the deacetylase co-factor PdcF is not

included in the schematic. **(D)** Comparison of IseA homologs from *B. subtilis*, *Ralstonia pickettii* and *B. anthracis*. The region on each protein that interacts with its cognate enzyme active site is colored dark blue. Structures of IseA-LytE (PDB: 8I2E) and Tai3:Tae3 (PDB: 4HZ9) are shown on the right. Only the catalytic domains of LytE and Tae3 are depicted.

example, genes that encode coat and exosporium proteins are largely absent from our dataset. Most of these proteins are not critical for heat-resistance but protect spores from enzymatic degradation of the spore peptidoglycan or function in adhesion [4,21]. Several of these factors have been identified and characterized based on homology and through proteomic studies [66]. The further characterization of these proteins and others will benefit from the ordered Tn collection.

Earlier work used a transposon site hybridization (TraSH) assay to screen for *B. anthracis* genes required for growth, sporulation, and germination [67]. This study identified 31 genes with *B. subtilis* homologs required for sporulation or germination, many of which were hits in our screen. Most of the other genes identified in the TraSH assay were not found in our Tn-seq screen, likely because mutants impaired in growth after germination were among the hits in the TraSH screen [67]. Nonetheless, this study was the first to suggest that *B. subtilis* is a worthy model for *B. anthracis*. Our results confirm and extend this conclusion. We have found that most of the core sporulation factors that drive morphogenesis in *B. subtilis* are required for endospore formation in *B. anthracis* and our cytological analysis of a subset of these factors suggest that they have similar functions. Several *B. subtilis* factors with more modest sporulation defects, like RacA that anchors the replication origins at the cell poles [68], SirA that inhibits additional rounds of replication [69,70], or FisB that catalyzes membrane fission [71] were not identified as *B. anthracis* sporulation genes in out screen. Directed experiments will be required to assess their contributions to morphogenesis in the pathogen. We also did not identify any of the sensor kinases that initiate the sporulation program in *B. subtilis*. The *B. subtilis* kinases have relatively modest phenotypes due to functional redundancy and pioneering work by Hoch and Perego indicates that the degree of redundancy among sporulation histidine kinases in *B. anthracis* is equally high [72,73].

Although most of the core sporulation machinery is shared between model and pathogen, our identification of >70 genes that are required for sporulation in *B. anthracis* but not *B. subtilis* points to meaningful differences in the sporulation program between the two species. Among the >50 genes that are present in both species but are only required for sporulation in *B. anthracis*, 25 are induced during sporulation in *B. subtilis* [16,30]. We suspect that the *B. subtills* homologs play similar roles in sporulation but their absence can be compensated for by other factors or pathways. The characterization of these genes in both *B. anthracis* and *B. subtilis* will provide insight into how and why these factors became more critical in the pathogen. It will also be interesting to investigate whether the remaining 25+ genes that are required in *B. anthracis* sporulation but not *B. subtilis* sporulation as well as the genes that are unique to *B. anthracis* are specifically induced during sporulation as was the case with *ipdA* (Fig 4E). Over two decades ago, DNA microarrays were used to profile genes that are induced during sporulation in *B. anthracis* [74]. These experiments defined five waves of gene expression during sporulation and identified >2000 genes that are growth phase-regulated. Some of the genes that we identified here were among this set. Transcriptional profiling using RNA-seq during sporulation in WT and in mutants lacking the sporulation transcription factors will supplement this earlier profiling study and will provide an important complement to the sporulation Tn-seq dataset described here.

It is noteworthy, that unlike *B. anthracis*, the molecular mechanisms underlying endospore formation in *Clostridiodes difficile* have striking differences in the factors and pathways that drive morphogenesis compared to *B. subtilis* [75–77]. Key differences include the absence of a phospho-relay that governs entry into sporulation in *B. subtilis* and likely *B. anthracis*; differences in cell-type-specific transcription factor activation throughout differentiation; and differences in the

factors involved in engulfment and cortex assembly. Finally, the germination pathway in *C. difficile* lacks the canonical germination receptors found in *B. subtilis* that function as ligand-gated ion channels. Instead, germinants are thought to be sensed by two soluble pseudoproteases CspA and CspC that, in turn, activate a subtilisin-like serine protease CspB. CspB proteolytically activates the cortex degrading enzyme SleC. Thus, germination in *C. difficile* proceeds from "outside in" rather than "inside out" in *B. subtilis* and *B. anthracis* [78]. Phylogenetic studies suggest that *B. subtilis* and *C. difficile* shared a common ancestor ~2.5 billion years ago [15,79] (compared to the ~800 million years between *B. subtilis* and *B. anthracis*). We suspect that the differences in sporulation in *C. difficile* are partly due to this deeper branching but also the differences in niches in which *B. subtilis* and *B. anthracis* reside and differentiate into dormant spores. Future studies into *C. difficile* and *B. anthracis* sporulation have the potential to reveal aspects of the morphogenetic pathways that are more plastic and those that are more hard-wired. Finally, the work described here provides a roadmap for similar genome-wide studies into endospore formation in species that reside between *C. difficile* and *B. subtilis* on the phylogenetic tree.

Finally, we turn to IpdA, a newly discovered *B. anthracis* sporulation factor that is not present in *B. subtilis*. IpdA is expressed at the onset of sporulation, prior to asymmetric cell division. Our genetic analysis and structural predictions suggest that IpdA inhibits the polysaccharide deacetylase PdaN and likely other PG deacetylase, enabling proper engulfment to occur (Fig 6A–6C). *B. anthracis* encodes 10 secreted polysaccharide deacetylase [61], several of which contribute to the high level of deacetylation of the *B. anthracis* peptidoglycan during growth [60]. More than 88% of the glucosamine residues and ~34% of the muramic acid residues lack N-acetyl groups in *B. anthracis* [64]. These modifications are associated with host lysozyme resistance and capsule attachment [80]. By contrast, during *B. subtilis* vegetative growth, only 17% of the glucosamine residues lack N-acetyl groups and virtually all muramic acid residues are N-acetylated [81]. Unsurprisingly, *B. subtilis* is highly susceptible to lysozyme (Fig Q*a* in S1 Text). Our data support the model that the IpdA-inhibition of PdaN and likely additional deacetylases identified in our AF-multimer screen ensure sufficient acetylated peptidoglycan to enable the cell wall hydrolase activity of SpoIID and/or SpoIIP that thins the septal PG and promote migration of the membranes around the forespore (Fig 6A–6C) [40,42,82]. Future studies will focus on testing the predictions of this model for IpdA function. More generally, our findings provide an example of how differences in the cell biology between *B. anthracis* and *B. subtilis* translate into distinct requirements for successful sporulation. We hypothesize that other pathogen-specific sporulation factors exist that remodel the cell envelope of vegetative *B. anthracis* as it enters sporulation. These factors could ensure the removal of *B. anthracis'* surface layer (S-layer) proteins, capsule, or neutral cell wall polysaccharides, none of which are present in the *B. subtilis* envelope.

Furthermore, our identification and characterization of IpdA highlight the potential for new discoveries that extend beyond the sporulation pathway. Specifically, our findings indicate that members of YkyA/PdcF protein family function as membrane tethers and co-factors for secreted enzymes like PdaN. Gene neighborhood analysis identified several examples of genes encoding YkyA/PdcF family members adjacent to secreted polysaccharide deacetylase genes (Fig S*a* in S1 Text). In all cases, AF3 predicted that these proteins interact in a manner similar to PdcF and PdaN (Fig S*b* in S1 Text). Furthermore, the Pfam database has several examples of fusions between a YkyA/PdcF domain and putative enzymatic domains (Fig T in S1 Text). Again, AlphaFold predicts these domains interact in a manner similar to PdcF and PdaN (Fig T in S1 Text). In addition, our characterization of IpdA led us to discover that IseA-family members can inhibit distinct enzymes involved in modifying the cell wall. As discussed above, *B. subtilis* IseA inhibits NlpC/P60 endopeptidases that cleave the peptide crosslinks of peptidoglycan. Another IseA-family member, Tai3, functions as an immunity factor of a type VI effector protein called Tae3 in *Ralstonia pickettii* (Fig 6D) [83]. Tae3 is a D,D-endopeptidase that lyses target cells by cleaving crosslinks in the peptidoglycan [84]. Tai3 prevents Tae3-mediated autolysis. Interestingly, all three IseA-family members inhibit their targets in different ways. IseA binds the catalytic groove of its target endopeptidases through an unstructured region that mimics the stem peptide (Fig 6D) [54]. Tia3 inhibits Tae3 using part of the same unstructured loop but binds its target in a distinct manner [83] (Fig 6D). Finally, IpdA is predicted to interact with PdaN using a short α-helix that lies adjacent to this unstructured region, making contacts with residues that surround the catalytic pocket (Figs 5D

and [6D]). We hypothesize that the IseA fold is readily evolvable to inhibit enzymes that cleave or modify extracytoplasmic glycopolymers. We suspect that IseA-family members inhibit a diverse collection of cell wall remodeling enzymes and that the IseA scaffold could be used to evolve novel and potent inhibitors.

## Conclusions

Our genome-wide analysis demonstrates that while *B. subtilis* provides an excellent framework for understanding sporulation, *B. anthracis* and likely other pathogens utilize distinct mechanisms with species-specific factors. Our work highlights how differences in cell envelope biology necessitate unique adaptations for successful spore formation. These findings underscore the dual themes of conservation and innovation that shape bacterial developmental programs. The datasets, tools, and conceptual framework we provide establish a foundation for dissecting sporulation in *B. anthracis* and we hope will catalyze comparative studies of other spore-forming Bacilli and Clostridial species.

## Materials and methods

### Bacterial strains and media

*B. anthracis* strains were cultured in brain-heart infusion (BHI) broth or agar. When required, BHI medium was supplemented with chloramphenicol (5 µg/mL), erythromycin (5 µg/mL), kanamycin (25 µg/mL), or spectinomycin (300 µg/mL). Sporulation of *B. anthracis* was induced at 37 °C by nutrient exhaustion in PA medium [85]. *B. subtilis* strains were grown in Luria–Bertani (LB) broth or agar. When required, LB medium was supplemented with chloramphenicol (5 µg/mL), erythromycin (1 µg/mL) plus lincomycin (25 µg/mL), or spectinomycin (100 µg/mL). Sporulation of *B. subtilis* was induced at 37 °C by nutrient exhaustion in supplemented Difco Sporulation medium (DSM) [86] or by resuspension according to the method of Sterlini–Mandelstam [87]. Sporulation efficiency was determined in 24–30-hour cultures as the total number of heat-resistant (*B. anthracis*: 65 °C for 30 min, or *B. subtilis*: 80 °C for 20 min) colony forming units (CFUs) compared to WT heat-resistant CFUs. The transposon-sequencing screen and the transposon-mutant collection were constructed in BaR1, the plasmid-free Sterne derivative of *B. anthracis* 9131 [88]. For sporulation efficiency assays, transposon mutants were transduced into BaR6 (*B. anthracis* Sterne NR-9401). *B. anthracis* 9131 (BaR1) spores are more adherent to glass and plastic tubes reducing the accuracy of CFU quantification. *B. subtilis* strains were derived from the prototrophic strain PY79 [89]. *B. subtilis* insertion-deletion mutants were from the *Bacillus* knock-out (BKE) collection [90] or were generated by direct transformation of plasmids into *B. subtilis*. All strains, plasmids, and oligonucleotide primers can be found in Tables A–C in S1 Text, and a description of plasmid constructions can be found in Supplemental Methods in S1 Text.

### CP51-mediated transduction

A temperature-sensitive mutant of the generalized transducing phage CP51 was used to transduce antibiotic-marked alleles [85]. Briefly, the donor strain was grown in BHI supplemented with 0.5% glycerol, then mixed with phage CP51 and PA soft agar. The mixture was poured onto nutrient broth/yeast extract/glycerol (NBYG) agar plates and incubated overnight at 30 °C. The transducing lysate was prepared by collecting the soft agar and resuspending it in BHI supplemented with 0.5% glycerol, 10% dimethyl sulfoxide, and 20 mM MgSO$_4$, followed by filtration through a 0.45 µm filter. The recipient strain was grown in BHI supplemented with 0.5% glycerol, mixed with the transducing lysate, and incubated with constant agitation at 37 °C for 45 min. The mixture was then plated on BHI agar plates supplemented with the appropriate antibiotic and incubated overnight at 37 °C. Transductants were confirmed by PCR.

### Electroporation

Transposon and allelic exchange plasmids, prepared from the *Escherichia coli* strain SC110 (*dam−*, *dcm−*), were transformed into *B. anthracis* by electroporation. Electrocompetent cells were prepared by growing *B. anthracis* in BHI with

0.1% glucose (BHIG) and 0.005 μg/mL tunicamycin. Low concentrations of tunicamycin increase transformation efficiency ~30-fold, likely due to the partial inhibition of TagO, the committing enzyme for the synthesis of secondary cell wall poly-saccharides [91,92]. Cells were harvested at an $OD_{600}$ of 0.3, chilled on ice, and washed 4 times in cold electroporation buffer (0.5 M sucrose, 10% glycerol). 100 μL of cells (concentrated 1000-fold) were mixed with ~2 μg plasmid DNA and transferred to a cold 1 mm electroporation cuvette. The electroporator settings were 25 μF, 200 Ω, 1.85 kV. Immediately after electroporation, cells were resuspended in 1 mL recovery medium (BHIG supplemented with 20 mM $MgCl_2$ and 2 mM $CaCl_2$) and incubated with constant agitation for 2 hour at 30 °C. The culture was plated on BHI agar supplemented with the appropriate antibiotic and incubated overnight at 30 °C.

### Allelic exchange

Deletions and integration of transgenes at neutral loci in the genome were generated by allelic exchange using pFR50, derived from pMAD [93]. This modified vector carries a thermosensitive origin of replication, an erythromycin resistance cassette for *Bacilli*, and mCherry fused to the *B. subtilis veg* promoter. 1 kb DNA fragments flanking the deletion or inser-tion site were subcloned into this vector with a second antibiotic resistance cassette between them. The plasmids were prepared in *E. coli* strain SC110 and electroporated into *B. anthracis* followed by selection on BHI agar supplemented with erythromycin. A streak-purified pink colony was then grown in 5 mL BHI-Erm at 30 °C. After 10 hour of growth, the culture was serially diluted and plated on BHI-Erm and incubated overnight at 42 °C to select for a single crossover integration (loop-in). A single colony was then grown without selection in 5 mL BHI at 30 °C for 10 hour. The culture was serially diluted and plated on BHI agar supplemented with the second antibiotic. Plates were incubated overnight at 37 °C. Colo-nies were screened for loop-out by the loss of pink color and sensitivity to erythromycin. Allelic exchange was confirmed by PCR.

### Transposon library construction and Transposon-insertion sequencing

Tn-seq was performed as described previously [17,24,25] with modifications described below. The *B. anthracis* trans-poson library was generated using the transposon delivery plasmid pFR38. This *E. coli-Bacillus* shuttle vector contains a temperature-sensitive replicon for *Bacilli*, an erythromycin resistance cassette, the mariner-*Himar1* transposase C9, and a spectinomycin resistance cassette flanked by inverted repeats recognized by the transposase. One of the two inverted repeats contains an MmeI endonuclease restriction site. There is enough readthrough of the *spec* gene to provide expression of downstream essential genes in operons when co-oriented. pFR38 was introduced into the *B. anthracis* WT strain BaR1 via electroporation. Transformants were selected at 30 °C on BHI agar supplemented with erythromycin. Ten test tubes with 4 mL of BHI supplemented with spectinomycin were each inoculated with 4 transfor-mants and grown at 22 °C for 16 hour. The cultures were pooled and transposants were selected on BHI agar plates supplemented with spectinomycin at 42 °C. Approximately 1,000,000 transposant colonies were pooled, diluted to an $OD_{600}$ of 5, distributed into 500 μL aliquots, and frozen in 15% glycerol at −80 °C. One aliquot was used to prepare genomic DNA (gDNA) to determine the depth of the library. The gDNA was digested with MmeI, followed by barcoded adaptor ligation. The transposon-chromosome junctions were amplified in 18 PCR cycles using a universal primer that binds the adaptor and a primer that binds within the transposon. The PCR product was gel-purified and sequenced on the Illumina HiSeq platform using TruSeq reagents (Tufts University Core Facility [TUCF]). The reads were mapped to the *B. anthracis* BaR1 genome.

### Gene essentiality

Essential genes were determined from the Tn-seq described above using EL-ARTIST [26]. 0.03 was used as the *p*-value cutoff for calling a region significantly underrepresented in reads and a sliding window size of 7 was used in the analysis.

## Sporulation Tn-seq screen

A frozen aliquot (500 μL) of the Tn-library was thawed, washed twice in PA medium, and diluted into 50 mL PA medium to an $OD_{600}$ of 0.05. The culture was grown at 37 °C and growth was monitored by $OD_{600}$ over time. An aliquot was harvested at the onset of starvation (input) and the rest of the culture was sporulated for 7 hour. Spores from the BaR1 strain are less adherent after 7 hour of sporulation and the maximum number of heat-resistant CFUs can be recovered. The sporulated sample was incubated at 65 °C for 30 min to kill vegetative cells and sporulation-defective mutants and plated on BHI agar. Approximately 1,000,000 heat-resistant colonies were pooled (output). gDNA was isolated from the input and output samples and processed for transposon-sequencing as described above. Reads were mapped to the *B. anthracis* BaR1 genome, tallied at each TA site, and genes in which reads were statistically underrepresented were identified using the Mann–Whitney *U* test. Visual inspection of transposon-insertion profiles was performed with the Sanger Artemis Genome Browser and Annotation tool [94]. Some of the genes listed in S2–S6 Tables that were chosen for further characterization based on visual inspection of the insertion profiles despite having a *P*-value >0.05 and/or a fold-enrichment of less than 3.

## Construction of the ordered transposon-mutant library by Knock-out Sudoku

The ordered library was constructed based on the protocols in Baym and colleagues 2016 [31]. Briefly, ~52,000 *B. anthracis* transposon-mutant colonies were picked using a RapidPick MP robot and transferred into 182 96-well plates containing 140 μL BHI-Spec per well. Construction of this progenitor collection was divided into three rounds with ~17,472 transposon mutants picked per round. The arrayed mutants were grown overnight at 30 °C with agitation at 600 rpm. The next morning, 4 × 20 μL of culture from each well was removed using a 96-channel Gilson PlateMaster P220 and deposited into four pools (rows, columns, plate rows, and plate columns) using 8-channel and 12-channel reservoir plates, and two single-well plates. In total, we generated 47 address pools: 8 row pools, 12 column pools, 13 plate rows, and 14 plate columns (using 13 × 14 plate grid). 50 μL of 20% glycerol was added to the remaining culture in each well of the progenitor collection and cryopreserved at −80 °C. The transposon-chromosome junctions of the transposons in each pool were determined by barcoded Tn-seq in a single NextSeq run.

To locate a Tn mutant in each gene within the progenitor collection, the sequencing dataset was parsed to identify the address pools in which each Tn mutant was found. A set of four pools containing a single Tn insertion dictates the coordinates of the well. In the cases where multiple distinct Tn insertion sites were identified for a given gene, we chose the Tn that had inserted co-directionally with the gene and had inserted closest to a position 1/5th of the gene length. When these criteria could not be satisfied, we used the next best Tn insertion (prioritizing co-directional insertion and then closeness to the 1/5th position). In total, we identified Tn mutants in 3,826 genes in singly occupied wells in the progenitor collection. We identified Tn mutants in 40 additional genes in doubly occupied wells. Tn mutants in 86 genes had insufficient address data to map. No mutants were identified for 1,504 genes in the *B. anthracis* genome. In many cases, this was due to gene essentiality and/or small size.

After transposon mapping, we used the colony picking robot to cherry-pick the Tn mutants from the progenitor collection and inoculate BHI supplemented with spectinomycin. These mutants were arrayed in 40 96-well plates. The 40 plate collection was pooled using the same strategy as the original progenitor collection with a 6 × 7 plate grid. The Tn-insertion sites in the 33 pools were determined using barcoded adaptors and Tn-sequencing. The insertions were mapped and their location in the condensed collection was validated. The nonredundant quality-controlled collection contains 3,826 Tn mutants that represent 70% of the 5,456 genes in the *B. anthracis* genome. We preserved two plates (D41 and D42) in which the 40 doubly occupied wells were struck for single colonies and single colonies inoculated into four wells in the final collection. Although these wells are not concretely mapped, they can be checked for the presence of either Tn mutant. The direction and position of the Tn insertions chosen for the final collection can be found in S8 Table.

The ordered *B. anthracis* Tn-mutant collection has been submitted to NIAID for consideration to be included in the BEI Resources program.

### Enrichment screen for Δ*ipdA* suppressors

Four colonies of Δ*04176*::*kan* (Δ*ipdA::kan*) in the BaR6 background were separately grown in 3 mL PA-Kan for 30 hour at 37 °C. The sporulated cultures were heat-killed (65 °C for 30 min), and 30 µL of the heat-treated material was used to inoculate 3 mL of PA-Kan medium for another round of sporulation. Heat-resistant CFUs were assessed after each cycle. After the fifth enrichment cycle, individual heat-resistant colonies from each culture were re-tested and gDNA from a representative suppressor from each culture was prepared and subjected to whole–genome sequencing (SeqCenter).

### Fluorescence and phase-contrast microscopy

Vegetative and sporulating cells and dormant spores were collected by centrifugation at 8,000 × *g* for 1 min and immobilized on 2% agarose pads. Phase-contrast and fluorescence microscopy were performed using an Olympus BX61 microscope equipped with an UplanF1 100× phase-contrast objective lens and a monochrome CoolSnapHQ digital camera (Photometrics). Membranes were stained with 1-(4-trimethylammoniumphenyl)-6-phenyl-1,3,5-hexatriene *p*-toluenesulfonate (TMA-DPH; Molecular Probes) at a final concentration of 50 µM. Exposure times were typically 100, 200, and 500 ms for YFP, CFP, and TMA-DPH, respectively. Image analysis and processing were performed using MetaMorph software (Molecular Devices; version 7.7).

### Immunoblot analysis

*B. anthracis* sporulating cells were collected by centrifugation at 16,000 × *g* for 2 min. Whole cell extracts were prepared by resuspending cell pellets in 50 µL lysis buffer (20 mM Tris pH7.5, 1 mM EDTA) containing 10 mg/mL lysozyme, 10 mM MgCl$_2$, 25 units DNase I, 0.08 mg/mL RNase A, and 1 mM phenylmethylsulfonyl fluoride) and incubating at 37 °C for 15 min. The samples were then mixed with an equal volume of 2× SDS-PAGE sample buffer containing 10% BME and heated at 50 °C for 15 min prior loading. Equivalent loading was based on OD$_{600}$ at the time of harvest. Proteins were separated by SDS-PAGE on a 20% polyacrylamide gel. Proteins were transferred to a PVDF membrane and blocked for 1 hour in 1× PBS-0.5% Tween-20 containing 5% nonfat milk. The blocked membrane was washed 4 times for 5 min with PBS-0.05% Tween-20 (PBST) and probed overnight at 4 °C with mouse anti-His tag antibody (GenScript) at a 1:4,000 dilution in PBST containing 3% BSA. The next morning the membrane was washed 4 times for 5 min with PBST and then incubated for 1 hour at room temperature with horseradish peroxidase-conjugated goat anti-mouse immunoglobulin G (Bio-Rad) at 1:10,000 dilution in PBST containing 3% BSA. The membrane was washed 4 times for 5 min with PBST and then incubated with Western Lightning reagent (PerkinElmer) and detected using a ChemiDoc MP, Imagin System (BIO-RAD). To control for loading, the lysates were separately resolved by SDS-PAGE and stained with Coomassie Brilliant Blue. Two biological replicates were analyzed, and a representative immunoblot and a Coomassie-stained gel are shown.

### AlphaFold2 Multimer screen

A one-by-all AF-M screen was performed using local Colabfold version 1.5.2 in January of 2025 [95,96]. IpdA (A0A6L-7GZZ9) was folded pairwise against the proteome of *B. anthracis* (UP000000594). Each prediction was run with all five AF-M models, using three recycles, no dropout, no templates. These runs were performed using a local installation of Colabfold running on NVIDIA A6000 GPUs managed by the HMS O2 computing cluster. MSAs were generated using MMseqs2 API server [97]. All predictions used paired and unpaired MSAs, and structures were unrelaxed. To analyze the results of the pairwise screen, we utilized a published analysis pipeline [58]. Predictions were considered a hit when the average models score was ≥3 and the average pAE ≤ 5 Å across the contact region in the best model. The metrics for each prediction are listed in S7 Table.

## Genomic neighborhood and domain fusion analysis

Gene neighborhood analysis was performed as previously described [98] using EFI-EST v2.0 from the Enzyme Function Initiative (available at: https://efi.igb.illinois.edu/efi-est/). Alignment score cutoff of 35% was used for the analysis. Gene neighborhood diagrams were generated to visualize the 10 nearest genes surrounding the *pdcF* (*ykyA*) gene. Domain fusions were identified for the YkyA protein family (Pfam ID: PF10368) on the InterPro website, available at: https://www.ebi.ac.uk/interpro/entry/pfam/PF10368/.

## Spot-dilution assays

*B. subtilis* strains were grown at 37 °C in LB medium until mid-exponential phase. Cultures were normalized to $OD_{600} = 0.6$ and 10-fold serial dilutions were generated. 5 µL of each dilution was spotted onto LB agar supplemented with or without indicated concentrations of IPTG. Plates were incubated at 37 °C overnight and photographed the next day.

## Lysozyme sensitivity assays

*B. anthracis* and *B. subtilis* strains were grown at 37 °C in sporulation media until mid-exponential phase and then back-diluted 50-fold and distributed into 96-well microtiter plates containing lysozyme. The $OD_{600}$ was recorded every 3 min for 5 hour using an Infinite M Plex plate reader (Tecan). The 96-well plate was maintained at 37 °C with constant agitation between measurements. Growth kinetics were analyzed in technical triplicate, and the values were averaged. The results reported are representative of experiments that were performed on at least two biological replicates.

## Supporting information

**S1 Data. Source data for Figs 1C, 4B, 5A, 5B, 5I and Figs Q*a*, Q*b*, and Q*c* in S1 Text.**
(XLSX)

**S1 Table. Gene essentiality in *B. anthracis* BaR1.**
(XLSX)

**S2 Table. Sporulation genes in *B. anthracis* with *spo* names.**
(XLSX)

**S3 Table. Sporulation genes in *B. anthracis* (lacking *spo* nomenclature).**
(XLSX)

**S4 Table. TCA cycle *B. anthracis* genes.**
(XLSX)

**S5 Table. Genes that are present in *B. subtilis* and *B. anthracis* but are only critical for sporulation in *B. anthracis*.**
(XLSX)

**S6 Table. Sporulation genes that are only present in *B. anthracis.***
(XLSX)

**S7 Table. Top hits from the AlphaFold-Multimer interaction screen of IpdA with all proteins in the *B. anthracis* BaR1 proteome.**
(XLSX)

**S8 Table. Location of the *B. anthracis* Tn mutants in the condensed collection.**
(XLSX)

**S1 Raw Images. Raw images related to Fig 5J.**
(PDF)

**S1 Text. Including: Supplemental Methods.** Plasmid constructions. **Table A.** List of *Bacillus* strains used in this study. **Table B.** List of plasmids used in this study. **Table C.** List of oligonucleotide primers used in this study. **Fig A.** Examples of genes required for *B. anthracis* sporulation that are also critical for *B. subtilis* sporulation. Transposon-insertion profiles from seven regions of the *B. anthracis* genome. Each contains a gene (or genes) that encode *B. subtilis* homologs that are required for *B. subtilis* sporulation. Each vertical line represents a transposon-insertion site, and its height reflects the relative number of insertions mapped to that site. The maximum height for each panel was set to the indicated number of reads shown in the right corner. Transposon-insertions in *B. anthracis sigF*, *spoIIAB* (*IIAB*), *spoIIAA* (*IIAA*), *spoIIID* (*IIID*), *spoIIQ*, *spoIIIE*, *spoIVA*, *spoIVFB*, *spoIVFA*, *spoVD*, and *spoVV* were underrepresented after sporulation, heat treatment, germination, and outgrowth compared to the input library. **Fig Ba.** Examples of transposon-insertion profiles of *B. anthracis* and *B. subtilis* genes that are critical for sporulation in *B. anthracis* but not in *B. subtilis*. Transposon-insertion profiles from three regions of the *B. anthracis* genome and the corresponding regions of the *B. subtilis* genome. Each line represents a transposon-insertion site, and its height reflects the relative number of insertions mapped to that site. The maximum height for each panel was set to 200 reads. Transposon-insertions in *yyaC*, *yqfT* (*T*), and *yhfN* were underrepresented after sporulation, heat treatment, germination, and outgrowth in *B. anthracis* compared to the input library (onset). By contrast, transposon-insertions in the homologous genes (dark gray) in *B. subtilis* were not underrepresented after sporulation, heat treatment, germination, and outgrowth. The *B. subtilis* Tn-seq data are from Meeske and colleagues 2016 [17]. **Fig Bb.** Examples of transposon-insertion profiles of *B. anthracis* and *B. subtilis* genes that are critical for sporulation in *B. anthracis* but not in *B. subtilis*. Transposon-insertion profiles from two regions of the *B. anthracis* genome and the corresponding regions of the *B. subtilis* genome. Each line represents a transposon-insertion site, and its height reflects the relative number of insertions mapped to that site. The maximum height for each panel was set to 200 reads. Transposon-insertions in *ytxC* and *ysxE* were underrepresented after sporulation, heat treatment, germination, and outgrowth in *B. anthracis* compared to the input library (onset). By contrast, transposon-insertions in the homologous genes (dark gray) in *B. subtilis* were not underrepresented after sporulation, heat treatment, germination, and outgrowth. The *B. subtilis* Tn-seq data are from Meeske and colleagues 2016 [17]. **Fig C.** Examples of *B. anthracis* sporulation genes that are not present in *B. subtilis*. Transposon-insertion profiles from two regions of the *B. anthracis* genome. Each line represents a transposon-insertion site, and its height reflects the relative number of insertions mapped to that site. The maximum height for each panel was set to the indicated number of reads shown in the right corner. Transposon-insertions in *01531* and *gerN* were underrepresented after sporulation, heat treatment, germination, and outgrowth in *B. anthracis* compared to the onset of starvation. **Fig D.** Conserved *B. anthracis* sporulation genes exhibit mutant phenotypes that are similar, but not identical, to *B. subtilis* mutants. Representative fluorescence images of the indicated *B. anthracis* mutants during a sporulation time course. Time after the onset of sporulation is indicated in the top left corner. All strains harbor a forespore (P$_{spoIIQ}$-*yfp*) transcriptional reporter. σ$^F$ (SigF) activity in the forespore was visualized with YFP (false-colored green). The membranes were stained with TMA-DPH (false-colored red in the merged images). The *B. anthracis spoIIE::Tn* mutant is delayed in polar division, fails to activate σ$^F$, and over time has impaired morphogenesis, similar to what has been reported for *B. subtilis* Δ*spoIIE*. The *B. anthracis spoIIR::Tn* mutant divides at both poles generating disporic sporangia. In some cases, both forespores activate σ$^F$, in others only one of the two forespores has σ$^F$ activity. Over time, additional polar septa form. The *spoIID::Tn* mutant is impaired in engulfment and, in most cases, fails to complete it. However, unlike the *B. subtilis* Δ*spoIID* mutant, the *B. anthracis* mutant does not form characteristic septal membrane bulges. The *spoIIQ::Tn* mutant engulfs almost as well as wild-type, but a subset of cells fails to complete the process. The mutant displays heterogeneous morphologies with unusual membrane blebs. The *spoIIIAE::Tn* mutant produces small forespores. This phenotype is reminiscent of the collapsed forespores observed in *B. subtilis* Δ*spoIIIA* and Δ*spoIIQ* mutants. Finally, the YFP fluorescence from the σ$^F$ reporter accumulates to wild-type levels at hour 3 of

sporulation but is almost completely lost by hour 4. The reduction in expression of a σF-responsive gene is consistent with previous studies in *B. subtilis* that report a loss in metabolic potential at late timepoints in the absence of the *spoIIIA* locus. Scale bars indicate 5 µm. **Fig E.** Conserved sporulation genes have similar but not identical mutant phenotypes. Representative fluorescence images of the indicated *B. anthracis* sporulation mutants during a sporulation time course. Time after the onset of sporulation is indicated in the top left corner. Membranes were stained with TMA-DPH. Carets highlight unusual features. Scale bar indicates 5 µm. **Fig Fa.** Genes that are important for sporulation in *B. anthracis* but are not critical for spore formation in *B. subtilis*. Representative fluorescence images of the indicated *B. anthracis* mutants at the indicated times after the onset of sporulation. All strains have forespore and mother cell transcriptional reporters. The membranes were visualized with TMA-DPH. Scale bar indicates 5 µm. **Fig Fb.** Genes that are important for sporulation in *B. anthracis* but are not critical for spore formation in *B. subtilis*. Representative fluorescence images of the indicated *B. anthracis* mutants at the indicated times after the onset of sporulation. All strains have forespore and mother cell transcriptional reporters. The membranes were visualized with TMA-DPH. Scale bar indicates 5 µm. **Fig G.** MurAB is important for sporulation in *B. anthracis*. (a) Representative phase-contrast and fluorescence images of wild-type (WT) and *murAB::Tn* strains during a sporulation time course. Time after the initiation of sporulation in PA medium is indicated in the top left corner. The membranes were stained with TMA-DPH. The sporulation efficiencies of wild-type and the mutant determined in the BaR6 background are shown in the bottom right of the phase-contrast images. (b) Representative fluorescence images of WT and *murAB::Tn* during exponential growth in BHI medium. Optical densities are indicated above the images. Exponentially growing cells lacking *murAB* are slighter wider and curvy or bent compared to WT. Scale bars indicate 5 µm. (c) Transposon-insertion profile of the region containing the *murAB* gene. Insertions in *murAB* are underrepresented after sporulation, heat treatment, germination, and outgrowth compared to the onset of sporulation. **Fig H.** Sporulating cells lacking *ipdA* and *sigE* lack septal bulges. Representative images of the indicated *B. anthracis* strains during a sporulation time course. Time after the onset of sporulation is indicated in the top left corner. In *B. subtilis,* sporulating cells lacking *sigE* do not initiate the mother cell program of gene expression, including expression of the engulfment cell wall hydrolases. *B. anthracis* sporulating cells lacking *ipdA* and *sigE* do not form septal bulges, suggesting the engulfment hydrolases are responsible for these bulges. Yellow carets highlight septal bulges in the Δ*ipdA* mutant. Membranes were stained with TMA-DPH. Scale bar indicates 5 µm. **Fig I.** IseA is not required for sporulation in *B. subtilis*. (a) Transposon-insertion profile of a region of the *B. subtilis* chromosome encompassing *iseA*. Transposon-insertions in *iseA* are not underrepresented after sporulation, heat treatment, germination, and outgrowth compared to the onset of starvation. The Tn-seq data are from Meeske and colleagues 2016 [17]. The *p*-value and fold-change in transposon-insertions after sporulation compared to onset are 0.10 and 0.93, respectively. (b) Representative images of the indicated *B. subtilis* strains during a sporulation time course. Time after the onset of sporulation is indicated in the top left corner. Sporulating cells lacking IseA engulf normally and are indistinguishable from wild-type. The Δ*spoIID* mutant displays characteristic septal bulges (yellow carets). The sporulation efficiencies of wild-type and the mutants are shown in the bottom right of the images at hour 4 of sporulation. Scale bar indicates 5 µm. **Fig J.** Transcription of *B. anthracis ipdA* but not *B. subtilis iseA* is induced at the onset of sporulation. Representative fluorescence images of *B. anthracis* and *B. subtilis* strains harboring *ipdA* and *iseA* promoter fusions to *yfp* or *venus* (false-colored green) during a sporulation time course. *B. anthracis* and *B. subtilis* were induced to sporulate by nutrient exhaustion and by resuspension, respectively. Membranes were stained with TMA-DPH. White carets highlight examples of sporulating cells that have induced P$_{ipdA}$-*yfp* prior to polar division. Scale bars indicate 5 µm. **Fig K.** AlphaFold3 predicts interactions between *B. subtilis* IseA and the NlpC/P60 catalytic domains of cell wall hydrolases. (a) Predicted structures of *B. subtilis* IseA (white) bound to LytE, LytF, CwlO, and CwlS (cyan). An unstructured loop in IseA is predicted to interact with the catalytic grooves of all four cell wall hydrolases. Only the catalytic domains of the four hydrolases are depicted. Catalytic residues are shown in magenta. Key interacting residues in IseA are colored dark blue. Predicted alignment error (pAE) plots, as well as predicted template modeling (pTM) and interface predicted template modeling (ipTM) scores, are shown for each prediction. (b) IpdA is not

predicted to interact with any of the NlpC/P60 domain-containing proteins in *B. anthracis*. pAE plots, and pTM and ipTM scores for each prediction are shown. **Fig L.** AlphaFold-predicted structures of *B. anthracis* IpdA (white) bound to four polysaccharide deacetylases (cyan). The catalytic residues of the deacetylase (magenta) and $Zn^{2+}$ ligand (purple) are shown. Predicted alignment error (pAE) plots, average model scores, and lowest pAE from the AlphaFold2-multimer screen are shown on the right. **Fig M.** Predicted structures of *B. anthracis* PdaN bound to PdcF with and without IpdA. (a) AlphaFold3-predicted complex of PdaN (cyan) and PdcF (pink). Cartoon and surface models are shown. The predicted lipidated N-terminus of PdcF is in dark blue. pAE plot and ipTM score are shown on the right. (b) AlphaFold3-predicted complex of PdaN (cyan), PdcF (pink), and IpdA (white). Cartoon and surface models are shown. pAE plot and ipTM score are shown on the right. **Fig N.** PdaN's N-terminus transitions from low to high per-residue local confidence in the presence of PdcF. AlphaFold3-predicted structures of PdaN, the PdaN/IpdA complex, and the PdaN/PdcF complex. The structures are colored based on a per-atom confidence estimates. Dark blue indicates very high-confidence, while yellow and orange indicate low and very low confidence, respectively. The per-residue local confidence of the N-terminus of PdaN is low or very low in the first two models. However, the per-residue local confidence of this region becomes high in the predicted complex with PdcF. The increase in local confidence provides additional support for the predicted PdaN/PdcF model. **Fig O.** The *pdaN* and *pdcF* genes are expressed prior to the onset of starvation and throughout sporulation. Representative fluorescence images of *B. anthracis* strains harboring *pdaN* and *pdcF* promoter fusions to *yfp* during a sporulation time course. Time (in hours) before and after the onset of sporulation are indicated in the top left corner. Membranes were stained with TMA-DPH. Scale bar indicates 5 µm. **Fig P.** Expression of *B. anthracis pdaN* and *pdcF* in *B. subtilis* inhibits cell elongation. (a) Photographs of spot-dilutions of the indicated *B. subtilis* strains on LB agar plates. 10-fold serial dilutions of the indicated strains were spotted on LB agar supplemented without or with 50 µM or 500 µM IPTG. The *pdaN* and *pdcF* genes were fused to the $P_{spank}$ ($P_{IPTG}$) promoter. The *B. subtilis pdaC* gene was fused to the $P_{hyperspank}$ ($P_{hyspank}$) promoter. The $P_{hyperspank}$ promoter is ~7-fold stronger than the $P_{spank}$ promoter. (b) Representative fluorescence and phase-contrast images of exponentially growing cells before and after addition of IPTG. Exponentially growing cultures of the indicated strains expressing cytoplasmic mCherry (red) were analyzed at the indicated times after IPTG (500 µM, final) addition. Short, chubby cells (yellow carets) and lysed cells (white carets) are characteristic of inhibition of cell wall elongation. Scale bar indicates 5 µm. **Fig Q.** *B. anthracis* cells lacking *pdcF* are more sensitive to lysozyme. Growth curves of *B. subtilis* (a) and *B. anthracis* (b) chronically exposed to lysozyme (2 µg/mL for *B. subtilis;* 600 µg/mL for *B. anthracis*). The indicated strains were grown in sporulation media until mid-exponential phase and back-diluted into fresh media with and without the indicated concentrations of lysozyme. $OD_{600}$ was recorded every 3 min for 5 h. Growth curves show the mean of three technical replicates ± standard deviation. (c) Growth curve of the indicated *B. anthracis* strains before and after acute exposure to 1 mg/mL lysozyme (arrow). The indicated strains were grown in sporulation media until mid-exponential phase and back-diluted to an $OD_{600}$ of 0.05. When the cultures reached an $OD_{600}$ ~0.3, they were divided in two. One subculture was exposed to lysozyme (1 mg/mL, final) and the other was left untreated. $OD_{600}$ was recorded every 30 min for 3 hour. (d) Representative phase-contrast images of the indicated strains in (c) 3 hour after addition of lysozyme (1 mg/mL, final). Yellow carets highlight lysed cells. Scale bar indicates 5 µm. Representative growth curves from one of two biological replicates are shown in panels a–c, with the corresponding underlying data available in S1 Data. **Fig R.** Expression of *B. anthracis pdaN* and *pdcF* in *B. subtilis* causes septal membrane bulging during sporulation. Representative images of the indicated strains during a sporulation time course. Sporulation was induced by the resuspension method, and IPTG (500 µM, final) was added to the cultures at the time of resuspension (0 hour). Cells were examined by fluorescence microscopy at the indicated time points. Septal bulges (yellow carets) are highlighted. Scale bar indicates 5 µm. Expression of *B. subtilis* PdaC, a membrane anchored MurNAc deacetylase that is not normally produced during sporulation, was included for comparison. **Fig S.** Genes encoding secreted polysaccharide deacetylases can be found adjacent to homologs of PdcF. (a) Gene neighborhood analysis using PdcF as the query identifies a subset of *Bacilli* in which genes encoding a putative polysaccharide deacetylase (*pda*) and a PdcF homolog (*pdcF*, *ykyA*) are

adjacent to each other. (b) AlphaFold3 predicts that the putative deacetylases (Pda) and their putative PdcF co-factors form a complex. Predicted structures are colored based on per-atom confidence estimates. Dark blue indicates very high-confidence, and light blue indicates confident. The pAE plots and ipTM and pTM scores are shown to the right. The AlphaFold3-predicted structures of the deacetylases are also shown. The predicted structures are colored based on per-atom confidence estimates. The N-terminal domains of both are orange indicating very low per-atom confidence estimates. **Fig T.** The YkyA domain is fused to different enzymatic domains. (a) Schematic from Pfam highlighting examples of stand-alone YkyA domains like PdcF and gene fusions with polysaccharide deacetylase domains and DUF4015 domains. (b) AlphaFold3-predicted structure of AWM75_04935 lipoprotein from *A. urinaehominis*. The structure on the left is colored based on a per-atom confidence estimates. Dark blue indicates very high-confidence, and light blue indicates confident. The two adjacent structures are colored using the same scheme as PdaN (cyan) and PdcF (pink). The YkyA domain is fused to the DUF4015 domain in this protein. The unstructured region that links the two domains interacts with the four-helix bundle, similar to PdaN and PdcF. The pAE plot is shown on the right.
(PDF)

## Acknowledgments

We thank all members of the Bernhardt–Rudner supergroup for helpful advice, discussions, and encouragement; the MicRoN core for advice on fluorescence microscopy, James Kirby, Steve Leppla, Terri Koehler, Roger Plaut, Anne Moir, David Popham, Michael Springer, Michael Baym, and Buz Barstow for sharing strains, protocols, instruments, and invaluable advice. A portion of this research was conducted on the O2 High Performance Computing Cluster, which is supported by the Research Computing Group at Harvard Medical School.

## Author contributions

**Conceptualization:** Fernando H. Ramírez-Guadiana, David Z. Rudner.

**Data curation:** Fernando H. Ramírez-Guadiana, Anna P. Brogan.

**Formal analysis:** Anna P. Brogan, David Z. Rudner.

**Funding acquisition:** David Z. Rudner.

**Investigation:** Fernando H. Ramírez-Guadiana, Anna P. Brogan, Yuanchen Yu, Caroline Midonet, Ian J. Roney.

**Methodology:** Fernando H. Ramírez-Guadiana, Ernst W. Schmid.

**Project administration:** David Z. Rudner.

**Software:** Joel W. Sher, Ernst W. Schmid.

**Supervision:** David Z. Rudner.

**Validation:** Fernando H. Ramírez-Guadiana.

**Writing – original draft:** Fernando H. Ramírez-Guadiana, David Z. Rudner.

**Writing – review & editing:** Anna P. Brogan, Yuanchen Yu, Caroline Midonet, Ernst W. Schmid.

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
