## [Editor Report · Decision Letter 0]

17 Sep 2025

Dear David,

Thank you for submitting your manuscript entitled "A high throughput genetic screen in Bacillus anthracis identifies similarities and differences in sporulation between the pathogen and the Bacillus subtilis model" for consideration as a Research Article by PLOS Biology.

Your manuscript has now been evaluated by the PLOS Biology editorial staff [as well as by an academic editor with relevant expertise] and I am writing to let you know that we would like to send your submission out for external peer review.

Once your full submission is complete, your paper will undergo a series of checks in preparation for peer review. After your manuscript has passed the checks it will be sent out for review. To provide the metadata for your submission, please Login to Editorial Manager (https://www.editorialmanager.com/pbiology) within two working days, i.e. by Sep 19 2025 11:59PM.

Kind regards,

Nonia

Nonia Pariente, PhD

Editor in Chief

PLOS Biology

on behalf of

Melissa

Melissa Vazquez Hernandez, Ph.D.

Associate Editor

PLOS Biology

---

## [Decision Letter · Decision Letter 1]

13 Oct 2025

Dear David,

Thank you for your patience while your manuscript "A high throughput genetic screen in Bacillus anthracis identifies similarities and differences in sporulation between the pathogen and the Bacillus subtilis model" was peer-reviewed at PLOS Biology. It has now been evaluated by the PLOS Biology editors, an Academic Editor with relevant expertise, and by three independent reviewers. As mentioned in my earlier email, I'm handling this temporarily while my colleague Melissa is out of the office.

You'll see that reviewer #1 is very positive about the study, and only has textual and presentational requests. Reviewer #2 calls this “an excellent contribution” and just has minor textual requests (their point 27 seems to request some data, but we will not require this). Reviewer #3 is also very positive, drawing parallels with your previous PLOS Biology paper on B. subtilis; he wants you to define “sporulation gene” and to flag some limitations/ caveats.

After discussion with the Academic Editor, I'd like to stress that no further experimental data will be required. Based on the reviews, we are likely to accept this manuscript for publication, provided you satisfactorily address the remaining points raised by the reviewers and the following data and other policy-related requests:

IMPORTANT - please attend to the following:

a) Please could you change your Title to something shorter and more direct? We suggest: "Identification of sporulation genes in Bacillus anthracis highlights similarities and significant differences with Bacillus subtilis"

b) Please address all of the requests from the reviewers.

c) Please address my Data Policy requests below; specifically, we need you to supply the numerical values underlying Figs 1ABC, 4B, 5ABH, S1, S2AB, S7C, S9A, S11AB, S12, S13AB, S16B, S17B, either as a supplementary data file or as a permanent DOI’d deposition.

d) Please cite the location of the data clearly in all relevant main and supplementary Figure legends, e.g. “The data underlying this Figure can be found in S1 Data” or “The data underlying this Figure can be found in https://zenodo.org/records/XXXXXXXX

e) Please make any custom code available, either as a supplementary file or as part of your data deposition.

f) Please include the URLs of your funder in the Financial Disclosure statement, and write out more explicitly who was funded by each grant.

We expect to receive your revised manuscript within two weeks.

*Published Peer Review History*

*Press*

Sincerely,

Roli

Roland G Roberts, PhD

Senior Editor

PLOS Biology

rroberts@plos.org

on behalf of

Melissa Vazquez Hernandez, Ph.D.

Associate Editor

PLOS Biology

DATA POLICY:

Regardless of the method selected, please ensure that you provide the individual numerical values that underlie the summary data displayed in the following figure panels as they are essential for readers to assess your analysis and to reproduce it: Figs 1ABC, 4B, 5ABH, S1, S2AB, S7C, S9A, S11AB, S12, S13AB, S16B, S17B. NOTE: the numerical data provided should include all replicates AND the way in which the plotted mean and errors were derived (it should not present only the mean/average values).

CODE POLICY

DATA NOT SHOWN?

REVIEWERS' COMMENTS:

Reviewer #1:

The manuscript by Ramírez-Guadiana et al. describes the use of TnSeq to identify genes required for spore formation, germination, and/or outgrowth (defined as sporulation for the purposes of the manuscript and this review) of the important pathogen Bacillus anthracis. Prior to this study, little was known about the genes required for sporulation in this pathogen or the related Bacillus cereus group because it had long been assumed that the sporulation program would be essentially the same in B. anthracis as in B. subtilis. However, as the authors point out, B. anthracis and B. subtilis diverged over 800 million years ago, and consistent with this phylogenetic divergence, they find that one-third of the genes important for sporulation in B. anthracis are not important in B. subtilis and that at least ~15% of the genes that modulate sporulation in B. anthracis are not conserved in B. subtilis. After constructing an ordered TnSeq library for B. anthracis, a notable feat in itself, the authors analyze the roles of several widely conserved sporulation genes, as well as new sporulation genes identified in their screen. Impressively, they delineate the mechanism of one of these novel sporulation genes by demonstrating that the gene encoding IpdA, which they name for "Inhibitor of Polysaccharide Deacetylases," directly binds to the polysaccharide deacetylase PdaN and likely antagonizes the activity of this peptidoglycan-modifying enzyme. Their data suggest that preventing the deacetylation of polar septum peptidoglycan promotes the activity of the peptidoglycan hydrolases responsible for driving engulfment. Excitingly, IpdA homologs are found in a range of organisms, with their phylogenetic analyses suggesting that the IpdA scaffold may generally inhibit enzymes that modify peptidoglycan. Collectively, their detailed mechanistic study of IpdA and comparative analysis of sporulation in B. anthracis and B. subtilis represent a major contribution to the sporulation field.

The manuscript is well-written, and the work presented has been conducted rigorously. I particularly appreciated the Discussion section, which is thought-provoking and effectively highlights the broader implications of their findings, such as the broader role of IseA homologs and how their comparative analyses may identify "hard-wired" vs. more "plastic" processes during sporulation. Finally, they propose that differences in cell envelope composition drives some of the organism-specific adaptations identified in their study.

I have only minor comments related to the presentation and discussion of the authors' results.

For the discussion of the results in Figure 3, I would suggest breaking up the paragraph into discrete paragraphs (line 331 after "spore"; line 336 after "discovered," and line 341 after "nutrients." It was a bit hard to keep track of all the phenotypes as written.

- I would also suggest reordering the microscopy images to match the order in which they are described in the text. Specifically, the yhfN mutant should move before the ysxE strain.

- Providing different-colored carets for the various phenotypes highlighted (e.g., circular spores vs. distorted forespores) would be helpful.

- The authors should provide scale bars on the images where they are zoomed in. Adding a small gap between the zoomed-in region of the dual reporter fluorescence microscopy image (between the third and fourth columns) would allow the reader to more quickly recognize that the 5.75 hr phase-contrast images are distinct from the fluorescence microscopy images shown to the left.

Do the authors know what the dark spots observed by phase-contrast microscopy in Figure 3 represent/ reflect?

Is the yyaC mutant more prone to lysis? It looks like some vegetative cells may already be lysing? Did the authors see a decrease in vegetative cell counts for any of the mutants analyzed?

Did the transposon used by the author have an outward-facing promoter? Would they expect their transposon insertions to have a polar effect on the expression of downstream genes? Please provide some of this information in the Methods and/or Results section.

Can the authors speculate on why the B. anthracis spoIIE mutant forms multiple septa relative to a B. subtilis spoIIE mutant.

For the membrane bulge observed in engulfment mutants, the authors should consider referencing the finding by Lopez-Garrido et al. (2018), which indicates that forespore shape is influenced by the translocation of the chromosome into the forespore.

- Is the membrane bulging in the B. anthracis spoIID mutant less frequent than some of the other mutants identified?

- Does a B. anthracis spoIIP mutant generate membrane bulges similar to the ipdA mutant (or are the bulges less frequent like in a spoIID mutant?)

Do the authors know whether PdaN or IpdA specifically localize to the polar septum? I am wondering if deacetylation is restricted to this site or whether all vegetative peptidoglycan is no longer deacetylated upon IpdA production.

Line 143: please hyphenate "sporulation-defective"

Line 269: consider adding a semi-colon after "engulfment" so that it reads "… engulfment,; however, unlike…"

Line 319: the phrasing "we liked the idea that…" might be a bit colloquial for a paper.

Line 332: "heterogeneous" has been misspelled.

Line 382: I think "investigated" should be "investigate"

Line 411: Please remove the "s" after "models"

Line 543: "anthracis" is misspelled

Figure 4 legend: "false" is misspelled

Reviewer #2:

In this study Ramirez-Guadiana et al. determined and analyzed the molecular basis of sporulation in Bacillus anthracis and compare these findings to what is known for the sporulation model organism Bacillus subtilis. The authors identified 159 putative sporulation genes by Tn-insertion analysis using NGS. Deeper analysis revealed 58 putative sporulation genes present in B. anthracis but not previously implicated in sporulation in B. subtilis. As an example, the authors then characterized one candidate B. anthracis protein termed IpdA encoded by the gene BA_04167 in detail. Thus, this study keeps the door open to similar explorations of functions of candidate genes/proteins left unattended herein.

General comments:

1) In my opinion, this study is an excellent contribution to our understanding of which genes/proteins are surely/likely/possibly involved in sporulation of the notorious biothreat agent B. anthracis. The strength of the approach described here is the combination of global methods such as transposon-mutagenesis and omics (NGS of Tn-mutants) as well as genetic manipulation (directed mutant analysis) followed by careful observation by fluorescence microscopy for phenotypic analysis and in silico structural biology approaches.

2) At the same time, the manuscript remains easy to read (though it puts a significant - positive - strain on the reader's stamina for its volume of text and data).

3) Overall, I encountered only minor issues related to writing or data presentation as outlined in detail below.

Specific (minor) comments:

4) Line 34: "new sporulation genes" is too vague. Better: "newly identified…in B. anthracis".

5) L 37: better "…deacetylase PdaN in B. anthracis".

6) L 45: better "B. anthracis endospores are the infectious biological entities that cause the zoonotic disease anthrax".

7) L 51-53: better "Environmental fluctuations possibly due to global climate change, or human activities such as deforestation…to the discovery of new…".

8) L 54-56: this section is very vague and concludes with a hollow statement not matching the quality of the rest of the manuscript text. I suggest the authors formulating a scientific rationale for studying sporulation in the high-consequence spore former B. anthracis. Compare L 118-121 for a well formulated statement.

9) L 78: "it" is vague, please use a specific sentence object such as "the mature spore" instead.

10) L 109 (and elsewhere): why only "~70% of the non-essential genes in B. anthracis"?

11) L 110: better: "…Tn mutations[sic!] that, in B. subtilis, are known to impair polar division…". Please check correct use of "mutant" and "mutation" throughout the text (for instance see L 220).

12) L 120: better "…and likely enable deeper…".

13) L 139: since "PA" has not been introduced at this point, better "…phage assay medium…".

14) L 151 (and elsewhere): does "at the onset of sporulation" essentially mean: before cells have initiated sporulation?

15) L 159 (and elsewhere): "(≤5% of wild-type)" does certainly always mean "(≤5% sporulation efficiency compared to wild-type) [L 176]", right? Then please complete the term where needed.

16) (L 177: there is a "the" missing before "cut off".)

17) L 222-225: what about the other 9 (i.e., 62 out of 71) hits?

18) (L 344: "…had strong sporulation defects. These mutants failed to mature…".)

19) L 370-375: for Fig. 4A-C the order of their first reference in the text is out of order.

20) (L 411: "An average model…")

21) L 452, 454 and 456: meant here instead: "mutant protein" or "variant protein".

22) (L 514: "…thought to be sensed…".

23) Fig. 5I is not mentioned in the text.

24) (L 592: "…derivative of B. anthracis…")

25) References: make sure to format the text (e.g., italicize species names).

26) Figure 2: add to caption the information that close up figures are close ups of the figures right above, respectively.

27) Figure 3: the percentages of sporulation given do not seem to match the visual observation of the images for ysxE::Tn and ytxC::Tn. The last two lines of the figure legend are descriptive results and should be restricted to the results text section.

Out of curiosity: the ysxE::Tn and ytxC::Tn mutants have low sporulation efficiencies but there are some spores. Have you repeatedly cultivated these mutants over several sporulation cycles to see if secondary mutations compensate for the Tn-induced defect?

28) Figure 4: in the figure legend, row 1 and 2 of panel A should be better explained. Lines 8 and 14-15, respectively, of the figure legend are descriptive results and should be restricted to the results text section.

29) Figure 5: lines 11-12 and 13-14, respectively, of the figure legend are descriptive results and should be restricted to the results text section.

30) Legends of supplementary figures: There are descriptive results here as well.

31) Figure S6B is missing a scale bar.

Reviewer #3:

[identifies himself as Patrick Eichenberger]

This article by Ramirez-Guadiana et al reports a Tn-seq screen aimed at identifying genes required for efficient sporulation in the bacterial pathogen Bacillus anthracis. It is a follow-up to a paper by the same group (Meeske et al 2016 PLOS Biol) that applied a similar technique to the study of spore formation in the model organism Bacillus subtilis. The study is impressive by its scope and thoroughness. One of the main conclusions is that there is a significant overlap between genes required for sporulation in both organisms, thus justifying the use of model organisms that are more amenable to genetic manipulation for the study of complex developmental processes. Nevertheless, the study also demonstrated that there are several species-specific differences that merit investigation directly in the pathogen. As proof-of-concept, the last section of the article focuses on the characterization of ipdA, a gene of previously unknown function that is necessary for sporulation in B. anthracis but absent in B. subtilis. In general, the experiments are very well done with reasonable interpretation of the data. The necessary controls were included. I have a few suggestions listed below, but my comments are mostly minor.

Main point:

I think it is important to define exactly what is meant by the term "sporulation gene" in the context of the paper as there are many potential definitions. The authors state that there are 159 putative sporulation genes in B. anthracis. However, this is based primarily on heat resistance, which is only one phenotypic category. I would argue that the number is in fact considerably higher than this, based on transcriptomics. A "sporulation gene" could also reasonably be defined as a gene expressed during sporulation, whose expression is dependent on a sporulation transcription factor. Furthermore, a large functional category of sporulation genes, those that encode coat and exosporium proteins, is left out of the total when heat resistance is the main criterion, because the spore coat is dispensable for heat resistance, but these genes are key for other functions. In fact, it is already known from previous work that the composition and morphology of the spore coat is one of the main differences between all species of endospore formers. However, it would not be easy to identify coat genes in a Tn-seq approach, because there is no simple way to screen for them as some of the phenotypes are subtle. Nevertheless, I think that it would be helpful to comment on that limitation in the paper.

Minor points:

l.81 "Although the mother" Should be "mother cell", not just "mother"

l.193 "Our Tn-seq screen identified 58 putative sporulation genes in B. anthracis that are present in B. subtilis but have not been reported to be required for efficient sporulation in the model (yellow circles in Fig. 1C and Table S5)." This is an oversimplification. While it is true that these genes were not identified in the B. subtilis Tn seq screen (2016 paper mentioned above), some of these genes had already been identified as genes required for sporulation in B. subtilis. For example, ypjB, yhbH and prkA are all featured as SigE-dependent genes that are necessary for heat-kill resistance in B. subtilis in Ref 16 (albeit with modest phenotypes). So, the statement that "25 of them are induced during B. subtilis sporulation but lack reportable phenotypes [16, 29]" is not a fully accurate representation of the previous work.

l. 236 "transcriptional reports" should be "transcriptional reporters"

l.251 "These pleiotropic phenotypes have not been reported for the B. subtilis spoIIE mutant and hint at differences between the two species." I would be more cautious here. The fact that they have not been reported does not mean that they don't exist in B. subtilis (in fact, I know from personal experience that a similar phenotype can be observed in a spoIIE mutant in B. subtilis after a few hours). It is fine to report the phenotype, but I would not infer that it signals major differences between the two species.

l.270 "mother membranes" should be "mother cell membranes".

l.345 "ysxE" You could mention here that YsxE has been characterized as an inner spore coat protein in B. subtilis (McKenney et al. 2010, Current Biology). Furthermore, since ysxE is part of the spoVID operon, it may play a role in a SpoVID checkpoint in B. anthracis similar to the mechanism described in B. subtilis by Delerue et al. (2022, Dev Cell).

Not sure that panel A in Fig.1 is necessary (could be moved to the suppl)

---

## [Editor Report · Decision Letter 2]

10 Nov 2025

Dear David,

Thank you for the submission of your revised Research Article "Identification of sporulation genes in Bacillus anthracis highlights similarities and significant differences with Bacillus subtilis" for publication in PLOS Biology. On behalf of my colleagues and the Academic Editor, Erin Danielle Goley, I am pleased to say that we can in principle accept your manuscript for publication, provided you address any remaining formatting and reporting issues. These will be detailed in an email you should receive within 2-3 business days from our colleagues in the journal operations team; no action is required from you until then. Please note that we will not be able to formally accept your manuscript and schedule it for publication until you have completed any requested changes.

IMPORTANT: Many thanks for the additional data strengthening the claims. The Academic Editor was able to evaluate this, but has a comment regarding the presentation of the curves in S17 "are the curves presented averages of replicates or a single representative curve? That should be denoted in the legend (average with SD would be ideal)". Could you please adjust this? I have asked my colleagues to include this request alongside their own.

PRESS

Sincerely, 

Melissa

Melissa Vazquez Hernandez, Ph.D., Ph.D.

Associate Editor

PLOS Biology
